# Deterministic formation of carbon-functionalized quantum emitters in hexagonal boron nitride

Manlin Luo[1,9], Junyu Ge[2,9], Pengru Huang [3,9], Yi Yu[1], In Cheol Seo [1,4], Kunze Lu [1,5], Hao Sun [3], Jian Kwang Tan [1], Beng Kang Tay [1,6], Sejeong Kim[7], Weibo Gao [1,5], Hong Li [2,6] ✉ & Donguk Nam [8] ✉

Forming single-photon emitters (SPEs) in insulating hexagonal boron nitride (hBN) has sparked wide interests in the quantum photonics. Despite significant progress, it remains challenging to deterministically create SPEs at precise locations with a specific type of element for creating defects. In this study, we present a straightforward approach to generate site-deterministic carbon-functionalized quantum emitters in hBN by harnessing ultrasonic nanoindentation. The obtained SPEs are high-quality and can be scaled up to large arrays in a single fabrication step. Comprehensive experimental analyses reveal that the insertion of carbon atoms into the hBN lattice is the source of the robust quantum emission. Complementary theoretical studies suggest possible candidates for the structural origin of the defects based on our experimental results. This rapid and scalable nanoindentation method provides a new way to create SPE arrays with specific types of atoms, enabling the comprehensive investigation of the origins and mechanics of SPE formations in two-dimensional (2D) materials and beyond.

In the rapidly evolving landscape of quantum technology, single-photon emitters (SPEs) stand out as fundamental components for quantum computing, secure communication, and precise sensing[1–6]. Notably, wide-bandgap materials such as diamond and silicon carbide have emerged as exceptional hosts for single-photon emitters[7–9]. They are distinguished by their exceptional optical properties and their ability to sustain stable quantum emission at room temperature, offering robustness and high fidelity in quantum operations[7,8,10–12]. However, despite the advantages of diamond and silicon carbide, the exploration of van der Waals materials introduces new possibilities,

enabling the integration of SPEs into two-dimensional (2D) quantum devices that are potentially more versatile for photonic applications. 2D materials offer enhanced fabrication flexibility, improved compatibility with various components, and cost-effective material options[13–16].

Particularly, hexagonal boron nitride (hBN), a 2D layered van der Waals crystal, has recently attracted much attention as a promising host for SPEs. This interest is due to its excellent mechanical and optical properties, and a substantial band gap of approximately 6 eV[17–23]. Structural defects in hBN have been attributed as probable

[1]School of Electrical and Electronic Engineering, Nanyang Technological University, Singapore, Singapore. [2]School of Mechanical and Aerospace Engineering, Nanyang Technological University, Singapore, Singapore. [3]Institute for Functional Intelligent Materials (I-FIM), National University of Singapore, Singapore, Singapore. [4]Quantum Innovation Centre (Q.InC) & National Metrology Centre (NMC), Agency for Science, Technology and Research (A*STAR), Singapore, Singapore. [5]Division of Physics and Applied Physics, School of Physical and Mathematical Sciences, Nanyang Technological University, Singapore, Singapore. [6]CINTRA CNRS/NTU/THALES, IRL 3288, Research Techno Plaza, Nanyang Technological University, Singapore, Singapore. [7]Department of Electrical and Computer Engineering, Sungkyunkwan University (SKKU), Suwon 16419, Republic of Korea. [8]Department of Mechanical Engineering, Korea Advanced Institute of Science and Technology (KAIST), Daejeon, Republic of Korea. [9]These authors contributed equally: Manlin Luo, Junyu Ge, Pengru Huang. ✉e-mail: ehongli@ntu.edu.sg; dwnam@kaist.ac.kr

origins of single-photon emission across the ultraviolet (UV) to the visible spectrum[24–28]. The outstanding optical properties including high brightness, photon emission rate, stability, and purity make hBN an ideal candidate for practical applications in various domains of quantum technologies[18,29–31].

The deterministic creation of defect-based SPEs at precise locations in hBN has been a long-standing goal, explored through only limited techniques so far, including nanoindentation with an atomic force microscope (AFM) tip[32], ion irradiation[33–36], electron beam[26,37], laser writing[29,38], patterned masks[39], and chemical vapor deposition (CVD) growth on nanopillars[40]. However, while these methods introduce structural defects with random atoms or vacancies, they lack the capability to selectively create defects with specific elemental compositions. Therefore, the origin of created defects in these ways have been hypothesized via experimental techniques and theoretical analysis, making it difficult to comprehensively investigate the actual origin. Researchers attempted to introduce specific atoms into hBN using techniques such as thermal annealing[41,42], ion implantation[43] and were able to come to a more robust conclusion of the origin of SPEs, such as carbon[43], in hBN. However, these methods did not allow the creation of SPEs at specific locations in a deterministic manner.

In this paper, we demonstrate the deterministic generation of an SPE array in hBN by physically inserting carbon atoms into hBN lattices through a straightforward process. The advanced ultrasonic nanoindentation technique[44] allows creating carbon-based defects at desired spatial locations with high precision. This scalable process is carried out at room temperature and is completed in 30 seconds, thereby reducing the energy requirements, which is a critical concern in the microfabrication industry[45]. The statistical experiment reveals that the activation rate of SPEs with carbon insertion is ~59%. By leveraging theoretical calculations in conjunction with our experimental findings, we identify potential candidates for the structural origin of the defects for SPEs. This work not only contributes to the fundamental understanding of defect formation in hBN but also opens new avenues for the controlled fabrication of high-quality SPEs, thereby advancing the development of integrated quantum photonics platforms.

## Results

### Inserting carbon atoms into hBN via nanoindentation

Figure 1 illustrates our approach employing deterministic ultrasonic nanoindentation to create an array of carbon-functionalized SPEs within multilayer hBN flakes. First, chromium tips of ~1 μm height are fabricated on a SiO₂/Si substrate (Fig. 1a) for nanoindentation. In this work, chromium is selected as the tip material for its high hardness[46], which ensures sharp and durable tips. Chromium is deposited onto polystyrene (PS) spheres, which serve as a mask, using the electron beam evaporator (see Methods and Supplementary Fig. S1). Figure 1b presents scanning electron microscope (SEM) images of the fabricated chromium tips, showing excellent uniformity of the sharp tip array. Subsequently, the substrate is sputtered with carbon, resulting in the tips coated with carbon atoms of 20-nm thickness (Fig. 1c). This carbon coating plays an important role in creating carbon-based defects in hBN, which will be explained in detail in the following discussion. For the comparison, we also prepared chromium tips without carbon coating. Bulk hBN crystals were exfoliated into layers and deposited onto a SiO₂/Si substrate. Subsequently, the substrates with carbon/chromium and chromium-only tips (tip substrates) are flipped to face the Si substrates with exfoliated hBN multilayers (hBN substrates). The two sets of stacked substrates are loaded into a customized ultrasonic nanoimprinter equipped with a sealed chamber (Fig. 1d), allowing the tips to indent the hBN layers under the vacuum condition[44]. Ultrasonic vibrations transmitted to the stacked chips result in the perforation of the hBN layers. As the tips with carbon coating break atomic bonds in the hBN lattices, carbon atoms from the surface of the tips are embedded in the hBN lattices. After removing the tip substrates, the hBN substrates are sent to an argon flow to remove excess carbon atoms not securely attached to the hBN lattices and are then annealed (See Methods). Images of the indented hBN flakes are provided in Supplementary Fig. S2. We observed significant fluctuations in emission intensity without the annealing, despite high brightness. However, after annealing, the emitters demonstrated reduced brightness but considerably enhanced stability (Supplementary Fig. S3, S4). This stabilization can be attributed to the annealing process, which not only recovers the hBN lattice damaged by nanoindentation but also reorganizes existing defects[37,47–49]. It is worth noting that our nanoindentation method has the potential to scale up the creation of SPEs in hBN across wafer sizes, since the chromium tip substrate can be fabricated to any required size, facilitating significant time and energy savings. The schematic illustration in Fig. 1e depicts possible scenarios for the indented hBN lattices with inserted carbon atoms, which will be explained in the theoretical modeling section. Upon the optical pumping, the defects are excited and produce single photons (See Methods).

### Characterization of indented hBN

To validate the hypothesis that carbon atoms can be inserted into the hBN lattices during nanoindentation, we employed Raman

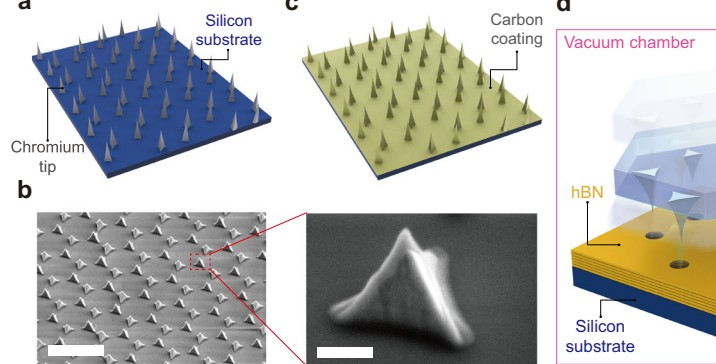
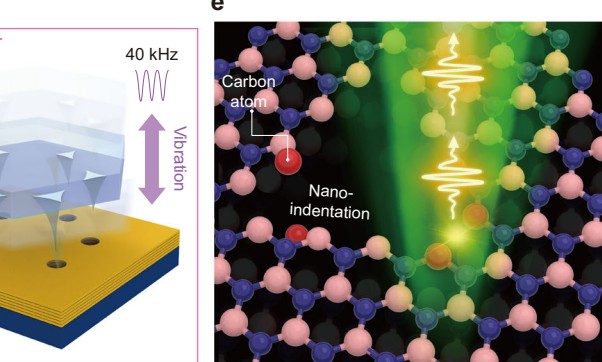

**Fig. 1 | Fabrication of carbon-enabled SPEs in hBN using ultrasonic nanoindentation. a** Schematic illustration of chromium tips on an SiO₂/Si substrate. **b** SEM images of the fabricated chromium tips. The left image shows a tilted view of the chromium tip array. Scale bar: 5 μm. The right image shows a zoomed-in view of the red dashed box, showing a single chromium tip. Scale bar: 500 nm. **c** Tips with carbon coating. Carbon atoms are deposited on chromium tips using sputtering. **d** Schematic illustration of nanoindentation step. The substrate with tips is flipped and stacked onto the substrate with exfoliated hBN layers. The stacked sample goes through ultrasonication in a vacuum chamber for nanoindentation. **e** Schematic illustration of the fabricated carbon-enabled SPEs in the hBN lattices, producing single photons upon optical pumping.

spectroscopy and 2D Raman scanning. Two samples are prepared for the characterization: the carbon sample indented by the carbon-coated tips and the control sample indented by the tips without carbon coating. Figure 2a compares the hBN Raman peak associated with the $E_{2g}$ phonon mode from nanoindentation sites in two samples, with dots for experimental data and solid lines for data fitted using the Lorentz function. The data derived from the carbon and control samples show Raman peaks at 1366.2 cm$^{-1}$ and 1368.4 cm$^{-1}$, respectively, showing a red-shift of 2.2 cm$^{-1}$ in the carbon sample. The full width at half-maximum (FWHM) of the Raman spectrum also increases from 7.6 cm$^{-1}$ in the control sample to 8.7 cm$^{-1}$ in the carbon sample, possibly due to the presence of carbon atoms in the hBN lattices[50]. To further demonstrate that the shift is related to the presence of carbon, we analyze the 2D mapping results of the position of the hBN Raman peak in the carbon and control samples (Fig. 2b and c). The locations of the nanoindentation sites are highlighted by dashed circles in both mappings. The hBN Raman peak positions of the indented sites in the carbon sample are clearly red-shifted, as evidenced by the bright spots in the mapping (Fig. 2b). The variability in the size and intensity of the bright spots in Fig. 2b may be attributed to the non-uniform density of the carbon atoms in indentation sites and is unlikely to be caused by strain. As evidenced by the Raman results in Supplementary Fig. S5, annealing effectively releases nanoindentation-induced strain, whereas wrinkle-related strain remains unchanged. In contrast, the nanoindentation without carbon coating results in no shift in the hBN Raman peak position (Fig. 2c). This comparison study supports the presence of carbon at nanoindentation sites.

The topological features of the indented hBN sample have been investigated using AFM. Figure 2d reveals a topography map showing an array of six nanoindentation sites. The inset displays the line scan across the white dashed line shown in Fig. 2d, indicating a hole diameter of 500 nm and a depth of approximately 47 nm. Notably, the hBN flake used in this study is approximately 80 nm thick, revealing that the nanoindentation penetrated about half of its thickness. The depth of the nanoindentations can be controlled by setting the sonication amplitude during the nanoindentation (See Methods). Supplementary Fig. S6 displays the AFM images and line profiles of various hBN nanoindentation sites. A flake thickness of ~80 nm and a nanoindentation size of ~700 nm without full piercing provides optimal conditions for SPE generation. This combination offers a more stable environment for defect stabilization and facilitates local lattice reconstruction upon nanoindentation, enabling deterministic creation of carbon-inserted SPEs in hBN.

To gain atomic-scale insight into the structure near nanoindentation sites, we employed aberration-corrected transmission electron microscope (AC-TEM) to investigate indented hBN flakes (See Supplementary Fig. S7). The high-resolution high-angle annular dark-field scanning transmission electron microscopy (HAADF-STEM) image in Fig. 2e shows a fracture edge formed by direct contact with the nanotip. The nanoindentation process does not yield well-defined geometries, and we observe regions of varying thickness along the edge (multilayer region 1, monolayer region 2, and open hole region 3). At the fractured edge, highlighted by red dashed lines, zig–zag-type edges dominate. Sharp corner-like geometries, and triangular vacancy structures adjacent to edges are also observed.

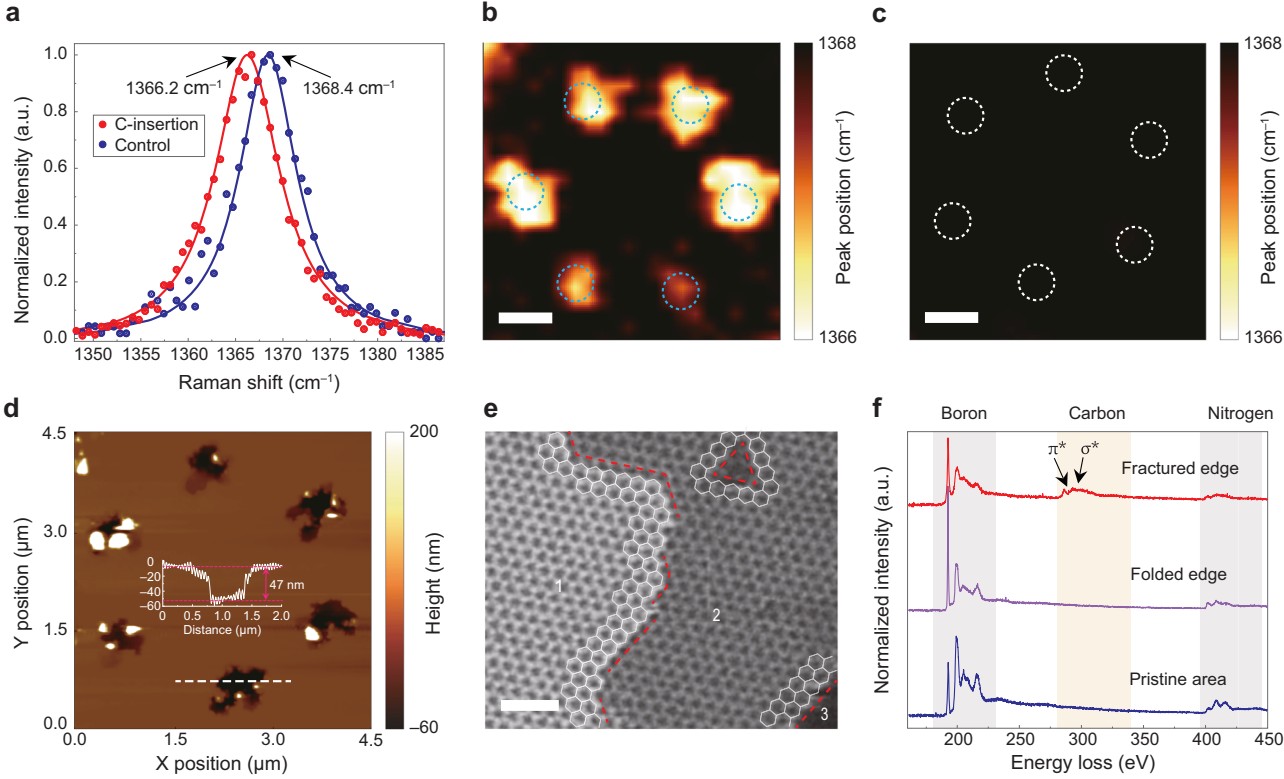

**Fig. 2 | Characterization of indented hBN layers with and without carbon insertion. a** Normalized Raman spectra measured from the carbon-inserted sample (red) and the control sample without carbon insertion (blue). Dots represent the measured data, and solid lines are Lorentzian fits to the experimental data. **b, c** Raman mapping results of hBN layers indented with carbon-coated tips (**b**) and tips without carbon coating (**c**). The dashed circles indicate the locations of nanoindentations. **d** Topographic map of an array of six nanoindentation sites measured by AFM. Inset: The 2D height profile of hBN across the white dashed line, which shows a ~47 nm variation. **e** High-resolution HAADF-STEM image of fractured edge region. Red dashed lines highlight the zig–zag-type edges. Labelled areas indicate: 1–multilayer region; 2–monolayer region; 3–open hole region. **f** EELS spectra acquired from 3 different regions near nanoindentation sites. Scale bars: (b–c) 1 μm, (e) 2 nm.

We used the electron energy-loss spectroscopy (EELS) detector to probe fracture edge (Fig. 2e, Supplementary Fig. S7d and f), pristine edge (Supplementary Fig. S7b) and folded edge (Supplementary Fig. S7c and e). This allowed us to examine the chemical composition of three different sites. The spectra are shown in Fig. 2f. As expected, the boron (-185 eV) and nitrogen (-400 eV) K-edges are observed in all regions. In contrast, a pronounced carbon K-edge signal (-285 eV) appears only at the fractured edge (red line). The carbon edge exhibits clear π* and σ* features, corresponding to transitions to unoccupied π and σ anti-bonding states, respectively, consistent with the presence of $sp^2$-hybridized carbon[51,52]. No carbon-related signals are detected in the spectra from the pristine hBN area (blue line) or folded edge (purple line), indicating that carbon incorporation is confined to regions that experienced direct mechanical contact with the carbon-coated tip. Atomic-scale contrast analysis for elemental identification is performed using the inverse fast Fourier transform (iFFT) image derived from Fig. 2f (Supplementary Fig. S7), providing evidence for carbon atoms incorporated at edge sites.

## Optical properties of carbon-functionalized emissions in hBN

To investigate the impact of carbon insertion on the optical properties of hBN, we conducted various experiments including photoluminescence (PL) and second-order autocorrelation measurements. Figure 3a and b present confocal PL maps of the carbon and control samples, respectively (see Methods and Supplementary Fig. S8 for more details on the confocal PL measurement). Insets display optical microscope images of the same area as in the PL maps. The samples

were illuminated with a 532-nm continuous-wave laser, and a filter set was used to collect the emission from 550 nm to 650 nm. In the carbon sample, bright emission is observed from all the nanoindentation sites (Fig. 3a). In contrast, the nanoindentation sites in the control sample do not show any PL emission within the wavelength range between 550 nm and 650 nm (Fig. 3b). Figure 3c shows emission spectra from nanoindentation sites in the carbon sample (white circle in Fig. 3a) and in the control sample. Upon optical pumping, the dominant zero-phonon-line (ZPL) emission emerges at 560.3 nm (2212.8 meV) for the carbon sample. In contrast, the spectrum from the control sample exhibits no visible peaks. The inset to Fig. 3c shows the zoomed-in spectrum of the ZPL peak with a FWHM of ~1.7 nm. The ZPL emission is accompanied by low-energy and high-energy phonon sidebands (PSB), which are attributed to longitudinal acoustic and longitudinal optical modes, respectively[37,53,54]. Supplementary Fig. S9 presents additional PL spectra from various nanoindentation sites, each displaying similar features. Statistical study results for the ZPL peak positions and the FWHM values are shown in Supplementary Fig. S10. The emission peaks reside primarily between 560 nm and 590 nm, with the majority of the FWHM values being less than 1.5 nm, aligning with previously reported carbon-based hBN SPEs[43]. We also conducted time-resolved PL measurements to see the temporal stability of the emission peak from the selected nanoindentation site (Fig. 3d). No discernable blinking or fluctuations in the emission wavelength (spectral diffusion) were observed for almost 10 minutes, demonstrating excellent stability of the produced SPEs. Notably, this stability persists even under room-temperature conditions (see Supplementary Fig. S11). To

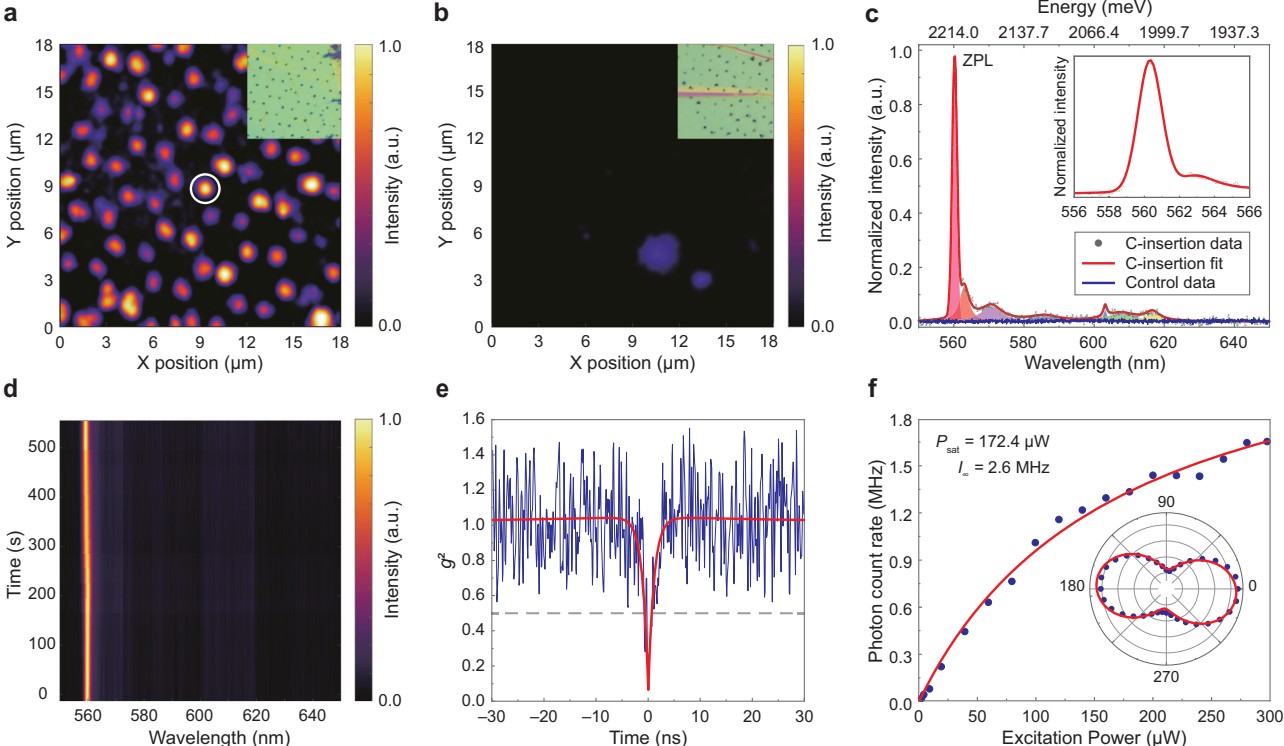

**Fig. 3 | Optical characterizations of carbon-functionalized emission in hBN layers. a, b** Scanning confocal PL maps of hBN layers indented with carbon-coated tips (**a**) and tips without carbon coating (**b**). Bright spots correspond to emission from carbon-functionalized defects. The spot highlighted by the white circle represents a representative emitter. Inset: Optical microscope images of the same area as the scanning confocal PL maps. **c** Normalized PL spectrum. Grey dots represent the measured data from the representative carbon-inserted emitter, while the red solid line is the fitted curve. Shaded regions indicate the areas integrated to extract the integrated PL intensity. The blue solid line displays the PL

spectrum from a control sample without carbon insertion. Inset: Close-up view of the ZPL. **d** Photoluminescence spectra as a function of time showing the photostability of the emitter. **e** Second-order autocorrelation function obtained from the single-photon emitter with continuous-wave excitation, demonstrating a $g^2(0) = 0.059 \pm 0.027$. The blue line represents the experimental data (without background correction), and the red curve shows the fitting result. **f** Fluorescence saturation curve obtained from the single defect, showing the saturate power of 172.4 μW. Inset: Excitation polarization recorded from the emitter. The solid red line is the fitting using a $\cos^2(\theta)$ function.

confirm the single-photon nature of bright emission from the nanoindentation sites in Fig. 3a, we performed second-order auto-correlation measurements for each bright site using a Hanbury Brown and Twiss (HBT) interferometry set-up. Figure 3e presents the auto-correlation function $g^2(\tau)$ of a representative emitter. By fitting the experimental $g^2(\tau)$ data using a three-level model, the resulting curve (red) dips to $0.059 \pm 0.027$ at zero delay time ($\tau = 0$), with the emission lifetime of the defect being ~1.1 ns, confirming anti-bunched photon statistics of single-photon emission. We note that the $g^2(\tau)$ data is not corrected for additional spectral filtering or the background subtraction, which may contribute to the small deviation at $g^2(0)$ from the ideal zero value. After checking all 68 bright sites in the carbon sample, emissions from 40 sites—approximately 59% of all nanoindentation sites—were confirmed as SPEs by measuring the second-order autocorrelation.

We measured the saturation behavior by increasing the excitation laser power to assess the SPE brightness. Considering the overall correction factors generated from our measurement (see more at Supplementary Note 2), Fig. 3f presents the emission intensity of SPE as a function of excitation laser power. The experimental data (blue dots) were fitted using a conventional saturated emitter model: $I = I_\infty \times P/(P + P_{sat})$, where $I_\infty$ and $P_{sat}$ represent the maximum achievable emission count rate and excitation power at the saturation excitation power, respectively. The fitted red curve reveals a $I_\infty = 2.64$ MHz at $P_{sat} = 172.4\ \mu W$.

The inset to Fig. 3f shows the excitation polarization plot, with corresponding fits obtained using a $\cos^2(\theta)$ fitting function. The

polarization visibility of the emitter is calculated with the maximum and minimum fitted intensity values of polarized emission, revealing a visibility of 55.75%. The low polarization visibility may result from the flake containing SPE becoming bent or warped due to vibrations incurred during the nanoindentation process. Therefore, we can conclude that the dipole direction of the SPE is not perpendicular to the optical axis[55]. Since the SPE dipole is not perpendicular, the collection efficiency may be reduced as some emissions escape outside the light cone of objective lens. Although the collection efficiency is not optimal due to the orientation of dipole is not perpendicular to the optical axis, this measured emission brightness ranks high among the SPEs synthesized through top-down fabrication techniques[32,36].

## Theoretical modeling

To ascertain the potential origin of the emission, we performed density functional theory (DFT) calculations on various atomic structures of carbon-inserted hBN via nanoindentations. A monolayer model with zig–zag-type edges was used (See Supplementary Fig. S12), as these edges are more stable[56] and commonly observed in our TEM analysis.

Since it is challenging to screen all possible defect structures, we focused on several representative configurations selected for their stability and likelihood. As previously demonstrated, the formation of SPEs is strongly correlated with carbon insertion. Representative carbon-related defect configurations are illustrated in Fig. 4a, including carbon atoms embedded at either boron-terminated (B-edge) or nitrogen-terminated (N-edge) zig–zag-type edges, labelled CC1–CC3.

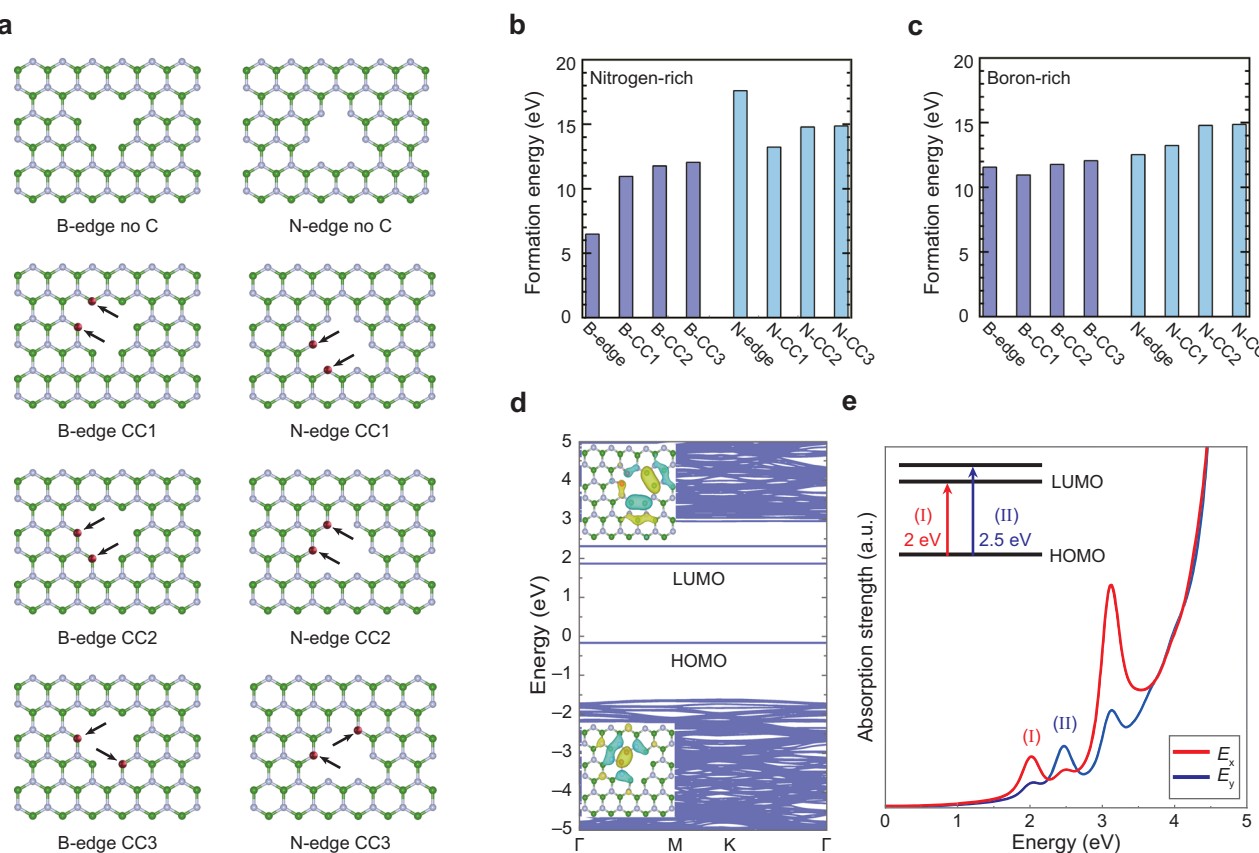

**Fig. 4 | Carbon dimer-functionalized model in hBN nanoindentation. a** Atomic structures of zig–zag edge with and without carbon atoms under boron-edge and nitrogen-edge terminations (color coding: boron in green; nitrogen in grey; carbon in red). **b, c** Calculated formation energies under (**b**) nitrogen-rich conditions and (**c**) boron-rich conditions. Darker blue bars denote boron-edge-based configurations, and lighter blue bars represent nitrogen-edge-based ones. **d** Calculated

electronic band structure for the carbon dimer-functionalized model. Insets: the calculated wave functions of the HOMO (bottom) and LUMO (top) states.
**e** Predicted absorption spectrum as a function of energy. The $E_x$ and $E_y$ denote the incident light polarizations aligned with the zig–zag and arm–chair edges, respectively.

To evaluate defect stability, we computed the formation energy as:

$$E_f = E(D) - E(hBN) + \sum_i n_i(\mu_i + E_i)$$

under nitrogen-rich and boron-rich conditions (Fig. 4b and c), representing two extreme cases of chemical potential during hBN growth or processing[57]. The results show that carbon atoms decorated at the boron-terminated edges forming C−N bonds are more stable than at nitrogen-terminated edges, in agreement with previous theoretical works[58,59]. Additionally, carbon dimers at edge or corner sites exhibit significantly lower formation energies than isolated carbon substitutions, likely due to local strain and under-coordination at geometrically confined sites. These combined insights on physical likelihood and energetic stability indicate that carbon impurities preferentially form dimer defects at the corner sites of boron-terminated zig–zag-type edge.

Figure 4d displays the calculated electronic band structure of the carbon dimer-functionalized SPE, revealing an energy gap of approximately 2.0 eV between highest occupied molecular orbital (HOMO) and lowest unoccupied molecular orbital (LUMO). This transition energy largely matches with the ZPL observed in our carbon-inserted samples[11,48]. For other defect candidates, the calculated HOMO−LUMO gaps are significantly larger or smaller than 2.0 eV (Supplementary Fig. S13), rendering them improbable contributors to the SPEs observed in our experiments. Insets in Fig. 4d depicts the wave functions of the HOMO and LUMO states for the proposed structure. Notably, the wave functions are highly localized at the carbon dimer site, indicating an on-site transition between defect levels. The HOMO states are mainly characterized by the bonding state, while the LUMO states are primarily contributed by the antibonding state of the carbon dimer. The on-site transition between the bonding and antibonding states as well as the matching energies between the ZPL and the theoretical value suggests that the carbon-dimer functionalized model is a very promising candidate for the SPEs observed in our experiments. Figure 4e presents the simulated absorption spectra for incident light polarized along the zig–zag ($E_x$) and arm–chair ($E_y$) edges, respectively, derived from the calculated electronic band structure. The absorption strength exhibits distinct differences between $E_x$ and $E_y$. There is a pronounced peak at 2 eV for $E_x$, labeled as (I) in the figure, which can be attributed to the transition from HOMO to LUMO. A second peak at 2.5 eV, labeled as (II), is associated with the transition from HOMO to a higher energy level. However, given that our excitation laser operates at a wavelength of 532 nm (2.3 eV), transitions like (II) and those at higher energies are not relevant to our analysis. These results validate the strong dipole transition between the bonding and antibonding states in the proposed carbon dimer-functionalized quantum emitters within the hBN nanoindentations.

## Discussion

In summary, we have demonstrated an effective platform for creating carbon-functionalized, deterministically positioned SPEs in hBN using ultrasonic nanoindentation techniques. Through carefully designed experiments and comprehensive characterizations, we confirmed the successful introduction of carbon atoms into hBN, enabling the scalable production of high-quality SPEs. Theoretical calculations provided valuable insights into the structural characteristics of the induced defects, complementing our experimental findings. The integration of these deterministic, high-quality SPEs into waveguides could lead to the development of scalable quantum photonic circuits[60]. Additionally, the application of strain engineering to achieve tunable single-photon emission may present exciting opportunities for tailored quantum light sources[61,62]. Also, this work present unique opportunities that the nanoindentation technique offers in the field of quantum technologies.

## Methods

### Sample fabrication

**Nanoindentation tips.** The nanoindentation tips are deposited with the aid of polystyrene (PS) spheres (Fig. S1). These PS spheres, with a diameter of 3.5 μm, are self-assembled onto a SiO$_2$ wafer by spin-coating into a monolayer that is both uniform and densely packed. Subsequently, chromium is uniformly deposited on the PS sphere-coated SiO$_2$ wafer through electron beam evaporation, resulting in the formation of sharply pointed tips. The height of chromium tips and the layout of their array are meticulously tailored by adjusting the diameter of the PS spheres used. Following the deposition process, annealing at 400 °C in an air environment is employed to remove the PS spheres and any excess chromium, leaving behind the nanotips adhered to the SiO$_2$ surface. These tips are then coated with carbon via sputtering, a process during which the thickness of the carbon layer is finely controlled by the duration of sputter coating. Control samples are prepared in a similar manner, with all steps identical except for the absence of the carbon sputtering step.

**hBN exfoliation.** hBN layers are then mechanically exfoliated from high-quality bulk material using the conventional scotch tape method. The exfoliated hBN flakes are transferred onto the SiO$_2$ substrate, with any residual adhesive removed by annealing in an air atmosphere at 400 °C for 30 minutes.

**Ultrasonic nanoindentation.** Ultrasonic nanoindentation is performed using Branson 2000X ultrasonic embossing equipment, operating at a frequency of 40 kHz. The process begins by stacking nanoindentation tips on hBN samples. This assembly is then positioned on the stage beneath the ultrasonic horn[44]. The ultrasonic nano-imprinter features a sealed chamber, allowing for the creation of either a vacuum or a rare gas environment. The nanoindentation process involves applying a 10%–20% amplitude and a force of 300–500 N for a duration of 20 seconds. Upon completion, the assembly is removed from the stage, and the tips are carefully detached to reveal the indented hBN samples.

**Annealing.** After removing the tip substrates, the hBN substrates are loaded to a CVD chamber. They are then subjected to an argon flow of 100 sccm for 1 hours, followed by annealing at 850 °C in an argon atmosphere with a flow of 50 sccm for 30 minutes. After that the samples are cooled down with a argon flow of 30 sccm overnight.

**TEM sample preparation.** hBN flakes were spin-coated with polymethyl methacrylate (PMMA) and subsequently detached from the silicon wafer by DI water. The released PMMA/hBN stack was then scooped onto a Quantifoil TEM grid. To remove the PMMA, the sample was exposed to acetone vapor followed by rinsing with isopropanol (IPA) and deionized water. Finally, the sample was dried on a hot plate at 50 °C.

### Structural characterization

**Surface topography characterization.** The morphological features and structural properties of the produced sample devices were examined using field-emission scanning electron microscopy (FESEM) (Apreo S) instrument and atomic force microscope (Park NX10). The electron acceleration voltage of FESEM was set to 5 kV. The AFM measurement was performed using a non-contact mode. The AFM images were corrected using XEI software (Park systems). HAADF-STEM images and EELS spectra were acquired using a JEOL JEM-ARM300F microscope equipped with double aberration correctors and a post-column Gatan EELS spectrometer. The system features an ultrahigh-vacuum STEM chamber and a cold field-emission gun, operated at 80 kV during our measurements.

## Optical characterization

**Raman spectroscopy and 2D mapping.** The Raman spectroscopy and 2D mapping were obtained with a WITec Raman system equipped with a 100× objective lens (numerical aperture of 0.9) and an 1800 lines/mm grating. The spatial resolution limit of this system is ~361 nm. A 532-nm laser, operating at a low power of less than 1 mW, was employed to illuminate or scan over the indented areas of the sample at room temperature.

**Confocal photoluminescence measurements.** Confocal photoluminescence (PL) measurements utilized a 532-nm excitation laser coupled with a 0.9 NA objective lens (ZEISS) (See Supplementary Fig. S4). The excitation laser was filtered through a 532-nm band-pass filter, followed by a linear polarizer and a quarter-wave plate to achieve circular polarization, followed by an additional linear polarizer. The emission was then sequentially filtered using a 550 nm long-pass filter, a 650 nm short-pass filter, and an 850 nm short-pass filter to refine the signal. The PL mapping was realized using dual-axis scanning Galvo Systems (Thorlabs). For cryogenic spectroscopy, the sample was cooled to 4 K by using a closed-loop cryostat (Montana Instruments). The PL spectra were recorded utilizing an Andor spectrometer featuring a 1200 lines/mm grating and a blade wavelength of 600 nm.

**Autocorrelation measurements.** The emission was split and sent into two fibre-coupled photon counting detector modules (Micro Photon Devices) in a Hanbury Brown and Twiss configuration. Photon correlation measurements were conducted using the PicoHarp time-correlated single-photon counting (TCSPC) system (PicoQuant).

## Theory calculation

**Density functional theory calculation.** Our calculations are based on density functional theory (DFT) using the PBE functional as implemented in the Vienna Ab Initio Simulation Package (VASP)[63,64]. The interaction between the valence electrons and ionic cores is described within the projector augmented (PAW) approach with a plane-wave energy cutoff of 500 eV[65]. Spin polarization was included for all the calculations. The calculations of pristine monolayer hBN were performed using a 50-atom 5×5 supercell, and the Brillouin zone was sampled using a (12×12×1) Monkhorst–Pack grid. The calculations for structures with edge defects were performed using a 10×10 supercell, which is sufficiently large to capture the localized nature of defects in the wide band gap of hBN. To minimize interactions between periodic images, a vacuum spacing of ~15 Å was applied along the non-periodic direction (Supplementary Fig. S12). Multilayer calculations show negligible variation in the HOMO−LUMO gap (Supplementary Fig. S14), justifying the monolayer approximation. In the structural energy minimization, the atomic coordinates are allowed to relax until the forces on all the atoms are less than 0.01 eV/Å. The energy tolerance is $10^{-6}$ eV.

## Data availability

The data that support the findings of this study are available within the article and its Supplementary Information. Any other relevant data are available from the corresponding authors upon request.

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

## Acknowledgements

The research of the project was in part supported by Ministry of Education Singapore (Award No: MOE-T2EP50221-0002 to H.L., MOE-T2EP50222-0018 to W.G.). H.S. is supported by Ministry of Education, Singapore, under its Research Centre of Excellence award to the Institute for Functional Intelligent Materials (I-FIM, project No. EDUNC-33-18-279-V12). This work was also supported by the TL seed project (TLSP25-04 to B.K.T., TLSP24-04 to W.G.). This work was supported by the National Research Foundation of Korea (NRF) grant funded by the Korea government (MSIT) (RS-2025-00520117 to D.N.). The authors would like to acknowledge and thank the Nanyang NanoFabrication Centre (N2FC) and the Facility for Analysis, Characterisation, Testing and Simulation (FACTS) at Nanyang Technological University, Singapore, for the use of their facilities.

## Author contributions

M.L., J.G., P.H., H.L. and D.N. initially conceived the initial idea of the project. Under the guidance of H.L. and D.N., M.L. and J.G. fabricated the samples. M.L. and J.G. performed and analyzed the atomic force microscopy (AFM) measurements. Guided by B.K.T., S.K., W.G. and D.N., M.L., J.G., Y.Y., I.C.S., K.L. and J.K.T conducted the optical measurements. P.H. and H.S. were responsible for performing the simulation and modeling. All authors analyzed and discussed the results. M.L., J.G., P.H., S.K., W.G., H.L. and D.N. contributed to writing and revising the manuscript.

## Competing interests

The authors declare no competing interests.
