## [Transparent Peer Review file · Nature Communications]

Deterministic formation of carbon-functionalized quantum emitters in hexagonal boron nitride

Corresponding Author: Professor Donguk Nam

Version 0:

Reviewer comments:

Reviewer #1

(Remarks to the Author)

Quantum emitters are crucial to quantum technologies, especially those based on photonics. In recent years, solid-state single-photon emitters (SPEs) from 2D materials like hexagonal boron nitride (hBN) have been extensively studied. However, their origins remain unclear, leading to challenges in their deterministic generation and may hinder their broader applications. The paper titled "Deterministic formation of carbon-functionalized quantum emitters in hexagonal boron nitride" by Luo et al. presents a timely and straightforward approach to investigating carbon-related defects, one of the promising potential origins. The manuscript is well-written, and the results are clearly presented. This research is of high significance to the community.

However, I have several concerns about the completeness of the experiments and simulations. Could the authors address the following points?

1. For the DFT simulations, how many layers of hBN were considered? As the authors said 'we developed a single-layer hBN model...' in main text, as well as "The monolayer of hBN and defects calculations were performed using a 50-atom 5x5 supercell." However, it later refers to multiple layers, stating, "A 15 Å vacuum space was used to avoid interaction between neighboring layers." Could the authors clarify this? Additionally, please discuss how the number of layers would affect the DFT results namely the energy level of the proposed carbon dimer defects.
2. The authors should discuss further possibilities or potential methods for obtaining direct evidence of defect origins. For instance, how likely is it to observe the defects under high-resolution TEM or AFM?
3. Regarding the method for carbon insertion and emitter generation, what is the limit of the created holes to form single emitters? Is there a relationship between the depth and creation efficiency? Is it possible to create emitters in a monolayer? Discussing these aspects, along with any challenges, would be helpful for future verification of defect types.
4. The author show experiments results of emitter stability in 10 mins(after post annealing), as the characterization was conducted at low temperatures in a vacuum condition. How about in ambient at room temperature?
5. The authors noted that before annealing, the emitters could be very bright but less stable. I suggest including at least one PL mapping in the supporting information of the unannealed sample post-nanoindentation but before Argon annealing, showing the emitters' location and brightness distributions.

Reviewer #2

(Remarks to the Author)

The manuscript by Luo et al. discusses single-photon emitters (SPE) given by defects in hexagonal boron nitride (hBN). SPE in hBN have been discussed for several years, and so far (i) it is unclear what the defects really are, and (ii) it remains unclear how they could be produced systematically and on purpose.

In the present manuscript the authors convincingly demonstrate that SPE can be generated by nanoindentation with carbon-decorated tips, with about half their samples resulting in SPE. The light emission is sharp and bright, and the SPE nature is documented by the autocorrelation function of the emission. Indentation without carbon yields no emitters, thus proving the crucial role of carbon. All of this looks promising.

However, three issues remain very vague and unclear (see below), so I do not really see progress towards systematic understanding of SPE fabrication:

(1) Fig. 3(a) shows that all dips become emitters after carbon-assisted indentation. As discussed in the main text, about 60 percent are SPE, indicating that in these cases only one defect is formed in the dip. From statistical consideration, a 60-percent probability of just having one defect in a dip should mean that there should be many cases with NO DEFECT AT ALL in a dip - in clear contradiction to Fig. 3(a). This is very puzzling.

(2) The manuscript does not really contribute to the puzzle of the nature of SPE in hBN (apart from the fact that in the present preparation technique, carbon is clearly involved). Apparently, some carbon-containing local defect(s) is/are formed - but where in the dip? The dips are several hundred nanometers wide and 50 nanometers deep, i.e. the hBN structure is severely deformed, probably with a lot of stress and strain, the significance of which is totally unclear. Plus, as mentioned above, I do not see why exactly one carbon-related defect should form in such a dip.

(3) The authors present some DFT-based calculations to support their findings, but I do find this very helpful, convincing, or conclusive: (i) The authors do not discuss how they "select the defects according to stability and likelihood"; (ii) The authors focus exclusively on defects in which small hBN flakes are cut out of the sample, with the edges then decorated by carbon atom(s), without stating why exactly such defects should form in the indentation process; (iii) DFT-derived spectra without many-body corrections on the single- and two-particle level are known to yield unreliable results, so I am afraid that the alleged matching with measured transition energies does not prove much.

One minor issue: Fig. 3(f) and the main text mention a saturation power of 172.4 microwatt, while the caption of Fig. 3 mentions 140.24 microwatt - maybe a misprint?

Version 1:

Reviewer comments:

Reviewer #1

(Remarks to the Author)

The authors have satisfactorily addressed my concerns regarding the completeness of the experimental characterisation. They have provided the required high-resolution TEM data, additional PL results on temperature and thickness dependence, as well as clarifications of the simulation settings. The results and methodology are sound. Given the significance of this work and its potential benefit to the solid-state single-photon community, I recommend publication in Nature Communications.

Reviewer #2

(Remarks to the Author)

Concerning the revised manuscript by Luo et al., I believe that many of the questions and concerns have been (at least partially) met and answered, and the manuscript has improved.

In particular,

(1) the authors have carefully considered and summarized the various possibilities of why and how defect formation may take place, and

(2) the authors have with significant care specified their calculations and discuss their merits and limits. The discussion of formation energies is nice and helpful.

I fully understand that the current approach is a significant step towards (semi-)systematic generation of carbon-related defects in boron nitride, as an "engineering" procedure, and therefore warrants publication.

I am still a little bit (personally) disappointed that a fully systematic understanding of the formation can still not be given in the manuscript, simply because it is not possible at this point. However, this is clearly not the fault of the authors but results from the complexity of the situation.

Reply to Reviewer #1

We sincerely thank the Reviewer #1 for the thoughtful and constructive feedback on our manuscript. The comments have been highly valuable in guiding us to improve the manuscript. In this letter, we carefully respond to all the concerns point-by-point, and we have revised the main text accordingly where appropriate. The changes made in the main manuscript are highlighted in red. For the added sections in the revised Supplementary Information, we highlight the section titles in red for reviewers to easily identify what sections have been added. We also submit a clean version without highlighted changes for reference.

Comment 0. Quantum emitters are crucial to quantum technologies, especially those based on photonics. In recent years, solid-state single-photon emitters (SPEs) from 2D materials like hexagonal boron nitride (hBN) have been extensively studied. However, their origins remain unclear, leading to challenges in their deterministic generation and may hinder their broader applications. The paper titled “Deterministic formation of carbon-functionalized quantum emitters in hexagonal boron nitride” by Luo et al. presents a timely and straightforward approach to investigating carbon-related defects, one of the promising potential origins. The manuscript is well-written, and the results are clearly presented. This research is of high significance to the community.

Response: We gratefully thank the Reviewer for the thoughtful and encouraging assessment of our work and manuscript. We are pleased that our approach was recognized as promising and straightforward. This comment led us to further reflect on key aspects of our study, which helped us improve and clarify the manuscript.

Comment 1. For the DFT simulations, how many layers of hBN were considered? As the authors said ‘we developed a single-layer hBN model...’ in main text, as well as “The monolayer of hBN and defects calculations were performed using a 50-atom 5×5

supercell.” However, it later refers to multiple layers, stating, “A 15 Å vacuum space was used to avoid interaction between neighboring layers.” Could the authors clarify this? Additionally, please discuss how the number of layers would affect the DFT results namely the energy level of the proposed carbon dimer defects.

Response: We thank the Reviewer for this comment. The Reviewer raised two important questions regarding our density functional theory (DFT) simulations. We first clarify the number of hBN layers used in our DFT computational model, with further details provided in **Sub-Section I**. In response to the Reviewer’s request to “*discuss how the number of layers would affect the DFT results namely the energy level of the proposed carbon dimer defects,*” we include additional analysis on the effect of interlayer interactions in **Sub-Section II**, where we examine how the number of layers influences the DFT results and the defect energy levels. Corresponding revisions have been made in both the main manuscript and the Supplementary Information, where appropriate.

Sub-Section I: We confirm that all DFT calculations were performed using a single atomic layer of hexagonal boron nitride (hBN) (monolayer). The confusion may have arisen due to our description of the vacuum spacing, which serves solely as a computational technique necessary for applying periodic boundary conditions along the out-of-plane (z) direction. To simulate an isolated two-dimensional (2D) sheet under periodic boundary conditions, we constructed a supercell containing only one atomic layer of hBN in the xy-plane and introduced a vacuum spacing of ~15 Å along the out-of-plane (z) direction. The total height of the supercell is 20 Å.

This model ensures negligible interaction between periodic images and accurately captures the physics of a monolayer system. Although the presence of vacuum along z direction may give the appearance of a vertically repeated structure, we emphasize that only one atomic layer is physically present in the model. A schematic representation of the supercell is provided in Fig. R1.

Figure R1 (Supplementary Figure S12 in revised SI) | Top and side views of the monolayer hBN supercell model used in DFT calculations. A supercell containing a monolayer of hBN with an inserted carbon defect is shown (atom code: boron: green; nitrogen: grey; carbon: red).

Sub-Section II: To address the Reviewer’s question regarding the influence of layer number on the DFT results, we performed additional calculations for the proposed carbon defect embedded in monolayer, bilayer, and few-layer hBN (Fig. R2a). In this simulation, we keep the vacuum spacing of ~ 15 Å to allow interaction between the hBN layers of interest. In each case, we calculated the energy gap between the highest occupied molecular orbital (HOMO) and lowest unoccupied molecular orbital (LUMO), which serves as a first-order approximation of the zero-phonon line (ZPL).

The results are presented in Fig. R2b. The HOMO–LUMO gap exhibits only minor variations (less than 0.1 eV) with increasing layer number, indicating the interlayer coupling has limited effect on the electronic structure of the localized defect. The defect orbitals remain spatially confined near the edge site, with little hybridization with adjacent layers. These results support our use of a monolayer model as a valid and computationally efficient representation for capturing the essential physics of the single-photon emitters (SPE).

Figure R2 (Supplementary Figure S14 in revised SI) | DFT-calculated HOMO–LUMO gaps of carbon-dimer defects in hBN. a, Atomic structures of the monolayer, bilayer, and tri-layer models. **b,** Calculated HOMO–LUMO gaps of the carbon-dimer defect structures in the monolayer, bilayer, and tri-layer models.

Action taken:

In the revised version of SI, we added Fig. R1 – 2 as Supplementary Fig. S12 and Fig. S14.

In the revised version, we added the following statements:

“A monolayer model with zig–zag-type edges was used (See Supplementary Fig. S12), as these edges are more stable and commonly observed in our TEM analysis.”

“The calculations of pristine monolayer hBN were performed using a 50-atom 5×5 supercell, and the Brillouin zone was sampled using a $(12\times 12\times 1)$ Monkhorst – Pack grid. The calculations for structures with edge defects were performed using a 10×10 supercell, which is sufficiently large to capture the localized nature of defects in the wide band gap of hBN. To minimize interactions between periodic images, a vacuum spacing of ~ 15 Å was applied along the non-periodic direction (Supplementary Fig. S12). Multilayer calculations show negligible variation in the HOMO–LUMO gap (Supplementary Fig. S14), justifying the monolayer approximation.”

Comment 2. The authors should discuss further possibilities or potential methods for obtaining direct evidence of defect origins. For instance, how likely is it to observe the defects under high-resolution TEM or AFM?

Response: We thank the Reviewer for this important suggestion. We fully agree that obtaining direct evidence for the atomic-scale origin of SPEs in 2D materials is of great significance. We also acknowledge that directly resolving the atomic structure of individual SPEs remains a significant experimental challenge^[1,2].

In our case, most nanoindentation-induced SPEs are located in non-perforated multilayer regions (i.e., regions that were indented but not pierced), which makes direct structural visualization of the emitter origin extremely difficult. The limited spatial resolution of our single-photon avalanche diode (SPAD)-based photoluminescence (PL) mapping (~200 nm) is comparable to the size of the nanoindentation features (~500 nm), making it challenging to pinpoint the exact location of the emitter within the indented region.

Nonetheless, we have taken several experimental steps to investigate the structural and chemical characteristics of the indented regions, as detailed in the following Sub-Sections: (i) atomic-scale imaging of nanoindentation boundaries, (ii) elemental identification of carbon insertion near indentation edges, (iii) control experiments to isolate the defect formation mechanism.

Sub-Section I: To gain atomic-scale insight into the structure near nanoindentation sites, we employed aberration-corrected transmission electron microscope (AC-TEM) to investigate indented hBN flakes that were transferred onto TEM grids.

Figure R3a shows a low-magnification TEM image of indented hBN transferred onto a Quantifoil TEM grid. The Quantifoil grid features a perforated carbon support film with regularly distributed circular holes, which offers both mechanical support and

open areas for high-resolution imaging of suspended regions. In our study, nanoindentations were uniformly introduced across the hBN flake, and all subsequent analyses were conducted only in the areas suspended over the holes to avoid any signal contributions from the underlying carbon film.

Figure R3b displays a high-angle annular dark-field scanning transmission electron microscopy (HAADF-STEM) image of a pristine hBN region, located a few nanometers away from the nanoindentation edge. The image reveals a well-preserved hexagonal lattice characteristic of hBN, confirming that the local crystalline structure remains intact at this distance from the mechanically modified area.

Figure R3c, e show a folded edge region formed during the nanoindentation process. The lattice remains continuous, with no visible signs of fracture or tearing. Although this edge is located at the indentation site, it did not come into direct contact with the carbon-coated tip. It serves as a reference region for subsequent analyses.

Figure R3d, f show a fracture edge formed by direct contact with the nanotip. The nanoindentation process does not yield well-defined geometries, and we observe regions of varying thickness along the edge (multilayer region 1, monolayer region 2, and open hole region 3). At the fractured edge, highlighted by red dashed lines in Fig. R3d, zig-zag-type edges are frequently observed. In addition, edge irregularities and triangular holes are found. These defect-rich sites are energetically favorable for stabilizing inserted carbon atoms^[3]. To capture the essential characteristics of such atomic-scale configurations, we used a triangular model in our simulations as a representative unit. This idealized structure allows systematic investigation of the stability and electronic levels of defects such as the carbon dimer at geometrically confined edge sites.

The three representative sites in the nanoindented hBN flake (Fig. R3b-d) provide different local environments. In the following sub-subsections, we will examine carbon insertion behavior at these sites in more detail.

Figure R3 (Figure 2 in the revised manuscript and Supplementary Figure S7 in the revised SI) | AC-TEM images of indented hBN. **a**, Low-magnification TEM image of indented hBN transferred onto a Quantifoil TEM grid. Scale bar: 2 μm . **b–d**, HAADF-STEM images of (b) a pristine hBN region, (c) a folded edge and (d) a fractured edge region. In **d**, red dashed lines highlight the zig-zag-type edges. **e–f**, Schematic illustrations corresponding to panels (c) and (d), respectively, highlighting (e) the multilayer folded edge and (f) the fractured edge. Red dashed boxes mark the regions shown in (c) and (d). Labelled areas indicate: 1 - multilayer region; 2 - monolayer region; 3 - open hole region. Scale bars: (a) 2 μm , (b,d) 1 nm, (c) 2 nm.

Sub-Section II: We used the electron energy-loss spectroscopy (EELS) detector to probe regions identified in Fig. R3b - d. This allowed us to examine the localized chemical composition of edge structures formed during nanoindentation.

The resulting spectra are shown in Fig. R4. As expected, the boron (~185 eV) and nitrogen (~400 eV) K-edges are observed in all regions. In contrast, a pronounced carbon K-edge signal (~285 eV) appears only at the fractured edge (red line, from Fig. R3d). The carbon edge exhibits clear π^* and σ^* features, corresponding to transitions to unoccupied π and σ anti-bonding states, respectively, consistent with the presence of sp^2 -hybridized carbon^[4,5]. No carbon-related signals are detected in the spectra from the pristine hBN area (blue line, from Fig. R3b) or folded edge (purple line, from Fig. R3c), indicating that carbon incorporation is confined to regions that experienced direct mechanical contact with the carbon-coated tip.

Figure R4 (Figure 2 in the revised manuscript) | EELS spectra acquired from 3 different regions near nanoindentation sites.

Sub-Section III: Control experiment reveals the necessity of carbon insertion during nanoindentation.

Figure R5. | Confocal PL mapping of carbon-sputtered hBN after bare-tip nanoindentation. Confocal PL mapping of the sample after bare-tip indentation followed by carbon sputtering and annealing shows no prominent emission from most indentation sites. The inset shows an optical microscope image of the indented hBN flake. The green dot is the laser spot.

To further understand the formation of carbon-related SPEs, a control experiment where carbon was introduced by carbon sputtering—not during the nanoindentation—was performed. Specifically, bare-tip indentation was first applied to hBN flakes, followed by sputtering of a uniform ~ 20 nm layer of amorphous carbon over the entire sample. The sample was then annealed at 850°C in argon.

As shown in the confocal PL mapping (Fig. R5), this control sample shows no clear emission features. There exist only a few very weak emission sites, primarily at locations with very large indentations, shown in the inset to Fig. R5. These regions are likely due to background PL from local accumulations of amorphous carbon, rather than defect-induced quantum emitters. Importantly, the majority of nanoindentation sites exhibited no detectable emission, indicating that post-deposition of carbon into nanoindentations, even followed by thermal activation, is insufficient to form stable SPEs.

These results support our conclusion that the insertion of carbon atoms must occur concurrently with the mechanical nanoindentation process, during the moment of lattice fracture, to enable successful formation of optically active defect states.

Action taken:

In the revised version of manuscript and SI, we added Fig. R3 - 4 as Fig. 2 and Supplementary Fig. S7.

In the revised version of manuscript, we added following:

“To gain atomic-scale insight into the structure near nanoindentation sites, we employed aberration-corrected transmission electron microscope (AC-TEM) to investigate indented hBN flakes (See Supplementary Fig. S7). The high-resolution high-angle annular dark-field scanning transmission electron microscopy (HAADF-STEM) image in Figure 2e shows a fracture edge formed by direct contact with the nanotip. The nanoindentation process does not yield well-defined geometries, and we observe regions of varying thickness along the edge (multilayer region 1, monolayer region 2, and open hole region 3). At the fractured edge, highlighted by red dashed lines, zig-zag-type edges dominate. Sharp corner-like geometries, and triangular vacancy structures adjacent to edges are also observed.

We used the electron energy-loss spectroscopy (EELS) detector to probe fracture edge (Fig. 2e, Supplementary Fig. S7d and f), pristine area (Supplementary Fig. S7b) and folded edge (Supplementary Fig. S7c and e). This allowed us to examine the chemical compositions of three different sites. The spectra are shown in Fig. 2f. As expected, the boron (~185 eV) and nitrogen (~400 eV) K-edges are observed in

all regions. In contrast, a pronounced carbon K-edge signal (~285 eV) appears only at the fractured edge (red line). The carbon edge exhibits clear π^* and σ^* features, corresponding to transitions to unoccupied π and σ anti-bonding states, respectively, consistent with the presence of sp^2 -hybridized carbon^{51,52}. No carbon-related signals are detected in the spectra from the pristine hBN area (blue line) or folded edge (purple line), indicating that carbon incorporation is confined to regions that experienced direct mechanical contact with the carbon-coated tip. Atomic-scale contrast analysis for elemental identification is performed using the inverse fast Fourier transform (iFFT) image derived from Fig. 2f (Supplementary Fig. S7), providing evidence for carbon atoms incorporated at edge sites.”

Reference for Comment #2 of Reviewer #1

1. Aharonovich, Igor, Dirk Englund, and Milos Toth. "Solid-state single-photon emitters." *Nature photonics* 10.10 (2016): 631-641.
2. Pellicciari, Jonathan, et al. "Elementary excitations of single-photon emitters in hexagonal boron nitride." *Nature Materials* 23.9 (2024): 1230-1236.
3. Huang, P., et al. "Carbon and vacancy centers in hexagonal boron nitride." *Physical Review B* 106.1 (2022): 014107.
4. Kim, Jaehyun, et al. "Electrochemically active porous carbon nanospheres prepared by inhibition of pyrolytic condensation of polymers." *Proceedings of the National Academy of Sciences* 120.19 (2023): e2222050120.
5. Toh, Chee-Tat, et al. "Synthesis and properties of free-standing monolayer amorphous carbon." *Nature* 577.7789 (2020): 199-203.

Comment 3. Regarding the method for carbon insertion and emitter generation, what is the limit of the created holes to form single emitters? Is there a relationship between the depth and creation efficiency? Is it possible to create emitters in a monolayer? Discussing these aspects, along with any challenges, would be helpful for future verification of defect types.

Response: We thank Reviewer for this important and forward-looking question. Understanding the influence of nanoindentation geometry and hBN thickness is crucial for clarifying the defect formation mechanism and guiding future SPE engineering. We summarize our findings into two Sub-Sections.

Sub-Section I: To investigate how nanoindentation size affects SPE formation, we focused on hBN flakes with a thickness of approximately 80 nm, where high-quality, high-yield SPE array are most frequently observed in our study.

We systematically compared the PL responses of nanoindentations with varying lateral dimensions with this thickness. Our observations suggest that there is a strong correlation between indentation size and SPE formation.

When the indentation diameter is significantly smaller than ~500 nm, we observed negligible signal under confocal PL mapping and autocorrelation measurements, suggesting that the limited contact prevents sufficient carbon insertion into the lattice. But for indentation diameters exceeding ~1 μm , we tend to observe multiple sharp peaks and a broad background in PL spectra, indicating the formation of multiple defects and excess amorphous carbon-related emission, respectively. A representative PL spectrum from such a large indentation is given in Fig. R6, which clearly reveals these multi-peak and strong background features. Our observation suggests that SPE generation tends to be most efficient when the nanoindentation size is near 700 nm.

Figure R6. | PL spectrum acquired from an indentation with diameter of ~ 1 μm .

Sub-Section II: To explore the effect of hBN thickness and indentation depth on the efficiency of SPE formation, we compare emitter generation in three distinct cases. The nanoindentation size was consistently controlled to ~700 nm in all three cases. As shown in Fig. R7, clear single-photon emission with a pronounced antibunching dip $g^2(0) < 0.5$ was only observed in flakes with a thickness of ~80 nm. In contrast, flakes that are either significantly thinner or thicker fail to produce observable SPEs.

For thin flakes (including monolayer hBN), we find that nanoindentation using sonication frequently punctures the flake. Notably, SPEs were not observed when the indentation pierced the flakes. This could arise from several factors:

- (i) As revealed by our TEM results, carbon insertion occurs only locally at the edges of a nanoindentation site created by the carbon-coated nanotip. In thinner flakes, the reduced edge area diminishes the probability of successful lattice insertion and defect formation.
- (ii) At the pierced geometries, generated defects are more vulnerable to perturbation from the substrate and surrounding dielectric environment,

which may destabilize or quench the formation of optically active quantum emitters.

For very thick flakes (e.g. ~ 200 nm), we also observe a lack of high-quality SPEs. We hypothesize that the increased mechanical rigidity of thicker flakes suppresses the localized strain concentration and lattice reconstruction required for SPE formation. As a result, nanoindentation may produce defect boundaries which are structurally rougher or spatially tilted. This reduces the likelihood of carbon atoms inserting into well-defined lattice positions to form optically active centers and may impair the efficient emission detection and coupling.

Figure R7. | Single-photon emission from hBN flakes of different thickness. Left: Autocorrelation result shows the antibunching. Right: Optical microscope images. Scale bar: $3.5 \mu\text{m}$.

Through this discussion, we have considered some physical factors that are crucial in creating optically active defect centers using our nanoindentation method. We found that a flake thickness of ~ 80 nm and a nanoindentation size of ~ 700 nm without full piercing provides optimal conditions for SPE generation. This combination offers a more stable environment for defect stabilization and facilitates local lattice reconstruction upon nanoindentation, enabling deterministic creation of carbon-

inserted SPEs in hBN. This analysis can provide valuable guidance for future efforts aimed at understanding and optimizing defect engineering strategies in 2D materials. Accordingly, we added the following sentence in the main text.

Action taken:

In the revised version of manuscript, we added the following:

“A flake thickness of ~80 nm and a nanoindentation size of ~700 nm without full piercing provides optimal conditions for SPE generation. This combination offers a more stable environment for defect stabilization and facilitates local lattice reconstruction upon nanoindentation, enabling deterministic creation of carbon-inserted SPEs in hBN.”

Comment 4. The author show experiments results of emitter stability in 10 mins (after post annealing), as the characterization was conducted at low temperatures in a vacuum condition. How about in ambient at room temperature?

Response: We thank the Reviewer for this important question. To evaluate the stability of the SPE under ambient conditions, we performed a systematic temperature-dependent study on a carbon-inserted annealed sample. A randomly selected SPE was characterized sequentially under four conditions: (1) 3.8 K in low vacuum, (2) 150 K in low vacuum, (3) 298 K (room temperature) in ambient air, and (4) back to 3.8 K in low vacuum. After each temperature change, the sample was allowed to stabilize for at least 2 hours prior to measurement.

To minimize the possibility of photobleaching and maintain emitter stability throughout the measurements, we used a relatively low excitation power (~5 μ W). Emitter stability was assessed by scanning confocal PL mapping, time-resolved PL spectroscopy, and second-order autocorrelation measurements under each condition.

Figure R8 (Supplementary Figure S11 in revised SI) | Confocal PL mapping of the same SPE at different temperatures. a–c, Normalized PL intensity maps measured at 3.8 K in low vacuum (a), 150 K in low vacuum (b), and 298 K in ambient air (c). All maps were normalized using the same intensity scale to enable direct comparison.

Figure R8 displays scanning confocal PL maps of the same SPE acquired under three temperatures: 3.8 K, 150 K and 298 K. The maps allow us to track the spatial position and brightness of the emitter.

As temperature increases, we observe a gradual decrease in peak intensity (from panel a to c), indicating reduced emission efficiency. This is likely due to enhanced non-radiative recombination at elevated temperatures. Notably, under room temperature ambient condition (Fig. R8c), the emission spot becomes slightly broader and more diffuse compared to the low-temperature cases. This broadening may result from increased phonon scattering at higher temperatures or ambient air effects.

Despite the reduced brightness and broadened shape, the SPE remains clearly visible and spatially stable throughout the temperature cycle, confirming its robustness.

Figure R9 (Supplementary Figure S11 in revised SI) | Temperature-dependent PL spectroscopy of a SPE. a, Time-resolved PL spectroscopy of the same emitter recorded sequentially under 3.8 K (low vacuum), 150 K (low vacuum), 298 K (ambient air), and back to 3.8 K (low vacuum). **b,** Normalized PL spectra extracted under each temperature condition.

The ZPL of the same SPE was monitored under each temperature condition for 500 s (Fig. R9a). At 3.8 K in vacuum, the emitter exhibits the highest optical stability, with consistent emission intensity and no discernible spectral diffusion. As the temperature increases to 150 K and 298 K (ambient air), the emitter remains active, though minor fluctuation in brightness and spectral sharpness become more apparent. No abrupt photobleaching was observed, although a moderate loss in brightness appeared near the end of measurement, indicating robust optical stability over an extended illumination period. In fact, the total excitation time across all measurements (including PL mapping and second-order autocorrelation function $g^2(\tau)$ tests) exceeds 2000s, during which the emitter remains reliable.

The corresponding PL spectra is shown in Fig. R9b. The emission peak position remains centered at ~584 nm across all temperatures. A minor blue shift (~0.5 nm) observed at 298 K, which may be attributed to temperature-induced spectral fluctuations and does not affect the overall spectral stability of the emitter. The full width at half maximum (FWHM) increases from 1.6 nm (3.8 K) to 2.1 nm (150 k), and 2.9 nm (298 k). This progressive broadening is consistent with phonon-induced dephasing and enhanced spectral diffusion. The intensity also decreases with increasing temperature, aligning with the confocal PL mapping results shown in Fig. R8. It is worth noting that the measurement data at 3.8 K, acquired after the full temperature cycling, exhibits largely unchanged temporal and spectral features.

Figure R10 (Supplementary Figure S11 in revised SI) | Second-order autocorrelation measurements. Second-order autocorrelation function of the same SPE measured at 3.8 K (low vacuum, bottom) and room temperature (ambient air, top).

Figure R10 presents the second-order autocorrelation function $g^2(\tau)$ measured at 3.8 K and 298 K. At cryogenic temperature, the emitter exhibits clear photon antibunching with $g^2(0) = 0.08$, indicating high-purity single-photon emission. Under room temperature and ambient conditions, the antibunching is still well below the threshold of 0.5, with $g^2(0) = 0.29$. The dip becomes shallower, reflecting increased background noise and reduced emission coherence. However, it continues to operate as an SPE at 298 K in air.

Action taken:

In the revised version of SI, we added Fig. R8 - 10 as Supplementary Fig. S11.

In the revised version of manuscript, we added the following:

“No discernable blinking or fluctuations in the emission wavelength (spectral diffusion) were observed for almost 10 minutes, demonstrating excellent stability of the produced SPEs. **Notably, this stability persists even under room-temperature conditions (see Supplementary Fig. S11).**”

Comment 5: The authors noted that before annealing, the emitters could be very bright but less stable. I suggest including at least one PL mapping in the supporting information of the unannealed sample post-nanoindentation but before Argon annealing, showing the emitters' location and brightness distributions.

Response: We thank the Reviewer for this constructive suggestion. We agree that including a confocal PL mapping of the unannealed sample after nanoindentation can provide helpful insights into the role of thermal stabilization. Such comparison enables the visualization of emitter properties before and after annealing in terms of both intensity and stability, helping to clarify the impact of excess surface carbon and lattice stabilization on SPE formation.

To address this, we performed a controlled comparative study on the same carbon-inserted sample, before and after the argon annealing. The sample was characterized using both (i) confocal PL mapping and (ii) real-time SPAD count monitoring. All measurements were conducted under identical conditions (laser power, scanning parameters, detector settings) to ensure comparability. The annealing was carried out at 850°C in an argon atmosphere with a flow of 50 sccm for 30 minutes.

Figure R11 (Supplementary Figure S3 in revised SI) | Confocal PL mapping of the same sample before and after annealing. a, PL map acquired after nanoindentation and carbon insertion, but before annealing. b, PL map of the same region after annealing

at 850°C in argon. Both maps are normalized to the same intensity scale for direct comparison.

Figure R11 displays the confocal PL maps of the same region (a) before and (b) after annealing. Both maps are normalized to a common intensity scale for direct comparison.

Prior to annealing (Fig. R11a), some highly luminescent points and strong background emission were observed, while the nanoindentation pattern appeared blurred and disordered. This is likely due to excess amorphous residues deposited on the hBN flake surface during nanoindentation process. These amorphous or weakly bound carbon species can generate broadband PL emission, even without being fully incorporated into the hBN lattice^[1]. Although optically bright, such emissions typically lack quantum characteristics. The absence of a clear antibunching signature with $g^2(0) < 0.5$ prior to annealing further supports this interpretation.

After annealing (Fig. R11b), the emission brightness at nanoindentations sites decreased slightly, but the nanoindentation sites became clearly identifiable with significantly reduced background noise. This improvement is due to the removal of excess carbon by the argon flow. Only after annealing do we observe emission centers that exhibit stable SPE with $g^2(0) < 0.5$, indicating the formation of well-incorporated defect states responsible for SPE emission.

Figure R12 (Supplementary Figure S4 in revised SI) | Real-time SPAD count monitoring of a selected emitter before and after annealing. Time-resolved single-

photon count rates were recorded at the same emitter location before (top, blue) and after (bottom, red) annealing. The histograms on the right display the corresponding distributions of count events over the same time window.

To further visualize the effect of annealing on emitter stability, we performed real-time SPAD count monitoring at a selected nanoindentation site before and after annealing, as shown in Fig. R12. Using an excitation power of 10 μW , the emitter exhibited a high photon count rate (up to 0.8 MHz) before annealing. But the signal was highly unstable with pronounced fluctuations and intermittent blinking behavior.

On the other hand, after annealing, the same site demonstrated a significantly reduced count rate (~ 0.2 MHz), but the emission became considerably stable over the entire measurement window. This result suggests that annealing not only removes excess carbon and filters out unstable emission, but also stabilizes the defect environment, enabling consistent photon emission.

The histograms on the right side of Figure R12 show the distribution of detected photon events across the same time window. Before annealing, the broad and skewed distribution reflects strong temporal fluctuations, while after annealing, the distribution narrows considerably, indicating improved temporal stability and reduced blinking.

Action taken:

In the revised version of SI, we added Figure R11 - 12 as Supplementary Fig. S3 - 4.

Reference for Comment #5 of Reviewer #1

1. Gehan, Rusli, A. J. Amaratunga, and S. R. P. Silva. "Photoluminescence in amorphous carbon thin films and its relation to the microscopic properties." *Thin Solid Films* 270.1 (1995): 160-164.

Reply to Reviewer #2

We sincerely thank the Reviewer #2 for the detailed and insightful comments on our manuscript. The comments have helped us refine the clarity of our work. In this following, we address each point raised by the Reviewer in a point-by-point manner, and we have revised the main text accordingly where appropriate. All changes made to the main manuscript are highlighted in red. For the added sections in the revised Supplementary Information, we highlight the section titles in red for reviewers to easily identify what sections have been added. We also submit a clean version without highlighted changes for reference.

Comment 0. The manuscript by Luo et al. discusses single-photon emitters (SPE) given by defects in hexagonal boron nitride (hBN). SPE in hBN have been discussed for several years, and so far (i) it is unclear what the defects really are, and (ii) it remains unclear how they could be produced systematically and on purpose.

In the present manuscript the authors convincingly demonstrate that SPE can be generated by nanoindentation with carbon-decorated tips, with about half their samples resulting in SPE. The light emission is sharp and bright, and the SPE nature is documented by the autocorrelation function of the emission. Indentation without carbon yields no emitters, thus proving the crucial role of carbon. All of this looks promising.

Response: We gratefully thank the Reviewer for the positive and encouraging comments regarding our work. We appreciate that the Reviewer recognized the potential significance of our study in addressing the long-standing challenges related to single-photon emitters (SPEs) in hexagonal boron nitride (hBN). We are encouraged that the Reviewer found our systematic approach based on nanoindentation with carbon insertion to be promising.

Comment 1: Fig. 3(a) shows that all dips become emitters after carbon-assisted indentation. As discussed in the main text, about 60 percent are SPE, indicating that in these cases only one defect is formed in the dip. From statistical consideration, a 60-percent probability of just having one defect in a dip should mean that there should be many cases with NO DEFECT AT ALL in a dip - in clear contradiction to Fig. 3(a). This is very puzzling.

Response: We thank Reviewer for this insightful observation. While it is true that Fig. 3a shows visible photoluminescence (PL) from almost all nanoindentation sites, this does not contradict our statement that ~60% of the dips produce SPEs.

We emphasize that SPE identification in our study relies on three combined criteria: (i) a localized PL bright spot, (ii) a single sharp emission line (ZPL), and (iii) $g^2(0) < 0.5$.

In Fig. 3a, each nanoindentation contains at least 1 carbon insertion. But ~60% of the sites host a single carbon-related structure that meets all 3 criteria for SPE identification, whereas the remaining ~40% contain either multiple emitters or none. All sites appear bright because they all contain carbon-related defects. There is no location that lacks a carbon defect or a detectable spectrum.

To elaborate on this point, we outline 6 possible scenarios to clarify the relationship between defects and PL emissions. For clarity, these are divided into three **Sub-Sections**.

Sub-Section I: Defects are present, but no emission is detected. In the following three scenarios, defects are indeed formed at nanoindentation/dip.

(i) **Not all defects are optically active:** Certain defect configurations may not support radiative recombination, either due to dominant non-radiative decay pathways or unfavorable electronic structure that suppress optical transitions.

(ii) **Excitation mismatch:** Some defect states may not be effectively excited by the excitation laser used.

(iii) **Detection window limitations:** Emission from certain defects may fall outside the spectral range of the detection window.

Sub-Section II: Emission is detected, but no SPE is confirmed.

(iv) **Non-SPE emission:** In some cases, one nanoindentation site exhibits multiple sharp ZPLs, but without clear photon antibunching behavior ($g^2(0) > 0.5$), suggesting the presence of multiple emitters. As shown in Fig. R13a, we experimentally observe such cases. Three narrow peaks are identified as ZPLs (ZPL1–3), along with broader spectral features between 610 nm and 650 nm that correspond to phonon sidebands (PSB) associated with these ZPLs. The presence of multiple ZPL–PSB pairs indicates that several independent SPEs coexist within one nanoindentation site, each contributing to the overall emission. Due to the absence of a clear antibunching signal, this nanoindentation emission site is not classified as a SPE.

Figure R13. | PL spectra revealing defect-related emission in hBN without SPE signature. a, PL spectrum showing multiple ZPLs and their corresponding PSBs. **b,** PL spectrum from carbon-related defect structures, where carbon contamination induces broadband emission.

(v) **Background luminescence.** Contamination-related signals may also contribute to visible PL without SPE signature. As shown in Fig. R13b,

the PL spectrum acquired from an indentation containing carbon-related defect structures exhibits a broad emission band^[1]. We performed a study on a carbon-inserted sample before and after the argon annealing.

Figure R11 (Supplementary Figure S3 in revised SI) | Confocal PL mapping of the same sample before and after annealing. a, PL map acquired after nanoindentation and carbon insertion, but before annealing. **b**, PL map of the same region after annealing at 850°C in argon. Both maps are normalized to the same intensity scale for direct comparison.

Figure R11 displays the confocal PL maps of the same region before (a) and after (b) annealing. Prior to annealing (Fig. R11a), some highly luminescent points and strong background emission were observed, while the nanoindentation pattern appeared blurred and disordered. This is likely due to excess carbon residues deposited on the hBN flake surface during nanoindentation process. These amorphous or weakly bound carbon species can generate broadband PL emission, even without being fully incorporated into the hBN lattice^[2]. Although optically bright, such emissions typically lack quantum characteristics. The absence of a clear antibunching signature with $g^2(0) < 0.5$ prior to annealing further supports this interpretation. After annealing (Fig. R11b), the emission brightness at nanoindentations sites decreased slightly, but the nanoindentation sites became clearly identifiable with significantly reduced background noise. This improvement is due to

the removal of excess carbon by the argon flow. Only after annealing do we observe emission centers that exhibit stable SPE with $g^2(0) < 0.5$, indicating the formation of well-incorporated defect states responsible for SPE emission.

Sub-Section III: Have confirmed SPE, but there are more than 1 SPEs on site. When other SPEs are filtered by the set-up, or not successfully excited.

Action taken:

In the revised version of SI, we added Figure R11 as Supplementary Fig. S3.

Reference for Comment #1 of Reviewer #2

1. Tang, Tsz Wing, et al. "Structured-Defect Engineering of Hexagonal Boron Nitride for Identified Visible Single-Photon Emitters." ACS nano 19.9 (2025): 8509-8519.
2. Gehan, Rusli, A. J. Amaratunga, and S. R. P. Silva. "Photoluminescence in amorphous carbon thin films and its relation to the microscopic properties." Thin Solid Films 270.1 (1995): 160-164.

Comment 2: The manuscript does not really contribute to the puzzle of the nature of SPE in hBN (apart from the fact that in the present preparation technique, carbon is clearly involved). Apparently, some carbon-containing local defect(s) is/are formed - but where in the dip? The dips are several hundred nanometers wide and 50 nanometers deep, i.e. the hBN structure is severely deformed, probably with a lot of stress and strain, the significance of which is totally unclear. Plus, as mentioned above, I do not see why exactly one carbon-related defect should form in such a dip.

Response: We thank Reviewer for this critical comment. We acknowledge that our present study does not quantify the exact number or precise location of carbon atoms inserted at each nanoindentation site, nor does it aim to fully resolve the atomic-scale identity of the defects. Our work is not a complete mechanistic determination of the nature of SPEs in hBN.

However, we believe it makes several valuable and important contributions toward this long-standing challenge. Specifically, we present an engineer approach that achieves ~60% success in SPE creation; we give optimal structural parameters that enable reproducible emitter generation.

We further correlate transmission electron microscopy (TEM) observations with emitter behavior to gain insights into defect formation near indentation edges; we exclude local strain as the dominant factor.

These findings provide engineering guidelines for reproducible SPE formation in 2D materials and help narrow the range of carbon-related defect configurations for future investigations.

Sub-Section I: To gain a deeper understanding of the SPE formation in hBN, we combined atomic-scale structural characterization using TEM (See Response to Comment 3) and systematically compared emitter generation across three contrasting cases. These complementary analyses allow us to build a clearer picture of the mechanisms governing and stabilizing the formation of carbon-related SPEs in hBN.

As shown in Fig. R7, the nanoindentation size was consistently controlled to ~700 nm in all three cases, but clear single-photon emission with a pronounced antibunching dip $g^2(0) < 0.5$ was only observed in flakes with a thickness of ~80 nm. In contrast, flakes that are either significantly thinner or thicker fail to produce observable SPEs.

For thin flakes (including monolayer hBN), we find that nanoindentation using sonication frequently punctures the flake. Notably, SPEs were never observed when the indentation pierced the flakes. This could arise from several factors:

(iii) As revealed by our TEM results, while the carbon-coated nanotip creates a nanoindentation site, carbon insertion occurs only locally at the edges. In thinner flakes, the reduced contact area further diminishes the probability of successful lattice insertion and defect formation.

(iv) At the pierced geometries, generated defects are more vulnerable to perturbation from the substrate and surrounding dielectric environment, which may destabilize or quench the formation of optically active quantum emitters.

For very thick flakes (e.g. ~200 nm), we also observe a lack of high-quality SPEs. We hypothesize that the increased mechanical rigidity of thicker flakes suppresses the localized strain concentration and lattice reconstruction required for SPE formation. As a result, nanoindentation may produce defect boundaries which are structurally rougher or spatially tilted. This reduces the likelihood of carbon atoms inserting into well-defined lattice positions to form optically active centers and may impair the efficient emission detection and coupling.

Through this discussion, we have considered some physical factors that are crucial in creating optically active defect centers using our nanoindentation method. We found that a flake thickness of ~80 nm and a nanoindentation size of ~700 nm without full piercing provides optimal conditions for SPE generation. This combination offers a more stable environment for defect stabilization and facilitates local lattice reconstruction upon nanoindentation, enabling deterministic creation of carbon-inserted SPEs in hBN. This analysis can provide valuable guidance for future efforts aimed at understanding and optimizing defect engineering strategies in 2D materials. Accordingly, we added one paragraph to the main text.

Figure R7. | Single-photon emission from hBN flakes of different thickness. Left: Autocorrelation result shows the antibunching. Right: Optical microscope images. Scale bar: 3.5 μm .

Sub-Section II: To evaluate the contribution of strain at nanoindentation sites, we conducted Raman spectroscopy mapping on hBN flakes indented by a bare tip before and after annealing. The goal is to assess whether strain persisted at the emitter sites and if annealing could relieve it.

Figure R14a shows the Raman 2D map of the E_{2g} phonon mode peak position, acquired before annealing. Red-shifted peaks are observed at the nanoindentation locations (marked by dashed circles), indicating local strain accumulation due to mechanical deformation^[1]. After annealing at 850°C in an argon atmosphere, the same region was mapped again (Fig. R14b). The Raman peak positions across the entire area, including the nanoindentations, shifted toward the peak position of unstrained hBN, indicating strain release during thermal annealing.

In contrast, Fig. R14c presents the Raman map of a region with naturally formed wrinkles, which are known to introduce strain in 2D materials. Unlike the nanoindented sites, the wrinkle-associated red shifts in Raman peak positions remained before and

after annealing, indicating that these are structurally different strain features not easily removed by thermal annealing.

Figure R14d compares normalized Raman spectra from four regions: nanoindentation sites before and after annealing, wrinkle and pristine areas. The red shift of the hBN peak at nanoindentations before annealing clearly disappears after annealing, while the wrinkle retains its red shift.

Figure R14 (Supplementary Figure S5 in revised SI) | Raman characterization of strain in hBN. a, Raman 2D map of peak positions of bare-tip-indented hBN before annealing. Dashed circles indicate nanoindentation positions. **b**, Corresponding Raman 2D map after annealing, acquired at the same location. **c**, Raman map of an hBN flake with wrinkles but no indentations. Dashed lines highlight the wrinkle features. **d**, Normalized Raman spectra extracted from the regions indicated in a–c: nanoindentation

sites before annealing (red), after annealing (dark blue), wrinkle (yellow) and pristine area (light blue). All measurements were performed on the same sample under identical experimental conditions. Scale bars: 1 μm .

Action taken:

In the revised version of manuscript, we added the following:

“A flake thickness of ~ 80 nm and a nanoindentation size of ~ 700 nm without full piercing provides optimal conditions for SPE generation. This combination offers a more stable environment for defect stabilization and facilitates local lattice reconstruction upon nanoindentation, enabling deterministic creation of carbon-inserted SPEs in hBN.”

In the revised version of SI, we added Figure R14 as Supplementary Fig. S5.

“The variability in the size and intensity of the bright spots in Fig. 2b may be attributed to the non-uniform density of the carbon atoms in indentation sites and is unlikely to be caused by strain. As evidenced by the Raman results in Supplementary Fig. S5, annealing effectively releases nanoindentation-induced strain, whereas wrinkle-related strain remains unchanged.”

Reference for Comment #2 of Reviewer #2

1. Chen, Xiang, et al. "Activated Single Photon Emitters And Enhanced Deep - Level Emissions in Hexagonal Boron Nitride Strain Crystal." *Advanced Functional Materials* 34.1 (2024): 2306128.

Comment 3: The authors present some DFT-based calculations to support their findings, but I do find this very helpful, convincing, or conclusive: (i) The authors do not discuss how they "select the defects according to stability and likelihood"; (ii) The authors focus exclusively on defects in which small hBN flakes are cut out of the sample, with the edges then decorated by carbon atom(s), without stating why exactly such defects should form in the indentation process; (iii) DFT-derived spectra without many-body corrections on the single- and two-particle level are known to yield unreliable results, so I am afraid that the alleged matching with measured transition energies does not prove much.

Response: We thank Reviewer for raising this important point. We agree that DFT calculations should be presented with clear justification regarding model assumptions, limitations, and interpretation. To make our response clearer and more logically structured, we have slightly reordered the sequence in which we address the Reviewer's questions. In the revised manuscript and below, we respond to the Reviewer's concerns in three Sub-Sections. **Sub-Section I** explains why we focus on edge-decorated carbon-related defects as the most relevant physical origin of observed SPEs. In **Sub-Section II**, we explain the rationale for the selection of defect models, including criteria related to their realistic likelihood and stability. In **Sub-Section III**, we discuss the known limitations of single-particle DFT and explain our rationale for using this approach in this study.

Sub-Section I: Our DFT models focus on edge-decorated carbon-related defects. To justify such configurations in the context of nanoindentation, logic is as below.

(i) (Main text) Spatial correlation between SPEs and nanoindentation sites: In the main text, we demonstrated that SPEs are located at sites corresponding to mechanical nanoindentation. The correspondence between indentation patterns and bright emission

centers clearly indicates that the formation of these emitters is directly linked to the nanoindentation process.

(ii) (Main text) Control experiments using carbon-free tips (main text): To isolate the role of carbon, we performed a control experiment using bare (no carbon coating) nanotips to create nanoindentations. The results showed no detectable PL emission. This comparison indicates that carbon is essential for SPE generation.

(iii) (Main text) Raman evidence of carbon insertion (main text): Raman spectroscopy revealed chemical modification attributable to carbon insertion during indentation.

(iv) (This response) Control experiment reveals the necessity of carbon insertion during nanoindentation.

Figure R5. | Confocal PL mapping of carbon-sputtered hBN after bare-tip nanoindentation. Confocal PL mapping of the sample after bare-tip indentation followed by carbon sputtering and annealing shows no prominent emission from most indentation sites. The inset shows an optical microscope image of the indented hBN flake. The green dot is the laser spot.

To further understand the formation of carbon-related SPEs, a control experiment where carbon was introduced after nanoindentation was performed. Specifically, bare-

tip indentation was first applied to hBN flakes, followed by uniform sputtering of a uniform ~20 nm layer of amorphous carbon over the entire sample. The sample was then annealed at 850°C in argon.

As shown in Fig. R5, only a few weak emission features were detected, primarily at locations with large indentations. These bright regions are likely due to background PL from local accumulations of amorphous carbon, rather than defect-induced quantum emitters. Importantly, the majority of nanoindentation sites exhibited no detectable emission, indicating that post-deposition of carbon into nanoindentations, even followed by thermal activation, is insufficient to form stable SPEs.

These results support our conclusion that the insertion of carbon atoms must occur concurrently with the mechanical nanoindentation process, during the moment of lattice fracture, to enable successful formation of optically active defect states.

(v) (This response) Direct atomic-scale imaging of nanoindentations: To gain atomic-scale insight into the structure near nanoindentation sites, we employed AC-TEM (JEOL JEM-ARM300F Grand ARM) to investigate indented hBN flakes that were transferred onto TEM grids.

Figure R3 (Figure 2 in the revised manuscript and Supplementary Figure S7 in the revised SI) | AC-TEM images of indented hBN. **a**, Low-magnification TEM image of indented hBN transferred onto a Quantifoil TEM grid. Scale bar: 2 μm . **b–d**, Atomic-resolution images of (b) a pristine hBN region, (c) a folded edge and (d) a fractured edge region. In **d**, red dashed lines highlight the zig-zag-type edges. **e–f**, Schematic illustrations corresponding to panels (c) and (d), respectively, highlighting (e) the multilayer folded edge and (f) the fractured edge. Red dashed boxes mark the regions shown in (c) and (d). Labelled areas indicate: 1 - multilayer region; 2 - monolayer region; 3 - open hole region. Scale bars: (a) 2 μm , (b,d) 1 nm, (c) 2 nm.

Figure R3a shows a low-magnification TEM image of indented hBN transferred onto a Quantifoil TEM grid. The Quantifoil grid features a perforated carbon support

film with regularly distributed circular holes, which offers both mechanical support and open areas for high-resolution imaging of suspended regions. In our study, nanoindentations were uniformly introduced across the hBN flake, and all subsequent analyses were confined to areas suspended over the holes to eliminate signal contributions from the underlying carbon film.

Figure R3b displays an atomic-resolution TEM image of a pristine hBN region, located a few nanometers away from the nanoindentation edge. The image reveals a well-preserved hexagonal lattice characteristic of hBN, confirming that the local crystalline structure remains intact at this distance from the mechanically modified area.

Figure R3c shows a folded edge region formed during the nanoindentation process. The lattice remains continuous, with no visible signs of fracture or tearing. Although this edge is located at the indentation site, it did not come into direct contact with the carbon-coated tip. Therefore, it serves as a reference region for subsequent analyses.

Figure R3d shows a fracture edge formed by direct contact with the nanotip. The nanoindentation process does not yield well-defined geometries, and we observe regions of varying thickness along the edge (multilayer region 1, monolayer region 2, and open hole region 3). At the fractured edge, highlighted by red dashed lines, zig-zag-type edges are frequently observed. In addition, edge irregularities and triangular holes are found. These defect-rich sites are energetically favorable for stabilizing inserted carbon atoms^[1]. To capture the essential characteristics of such atomic-scale configurations, we used a triangular model in our simulations as a representative unit. This idealized structure allows systematic investigation of the stability and electronic levels of defects such as the carbon dimer at geometrically confined edge sites.

We employed the embedded EELS detector to probe regions identified in Fig. R3b - d. This allowed us to examine the localized chemical composition of edge structures formed during nanoindentation.

The resulting spectra are shown in Fig. R4. As expected, the boron (~185 eV) and nitrogen (~400 eV) K-edges are observed in all regions. In contrast, a pronounced

carbon K-edge signal (~ 285 eV) appears only at the fractured edge (red line). The carbon edge exhibits clear π^* and σ^* features, corresponding to transitions to unoccupied π and σ anti-bonding states, respectively, consistent with the presence of sp^2 -hybridized carbon^[2,3]. No carbon-related signals are detected in the spectra from the pristine hBN area (blue line) or folded edge (purple line), indicating that carbon incorporation is confined to regions that experienced direct mechanical contact with the carbon-coated tip.

Figure R4 (Figure 2 in the revised manuscript) | EELS spectra acquired from 3 different regions near nanoindentation sites.

Together, these five sets of evidence establish a coherent and experimentally grounded rationale for focusing on edge-decorated carbon defect structures in our DFT analysis. The physical process of nanoindentation provides both the mechanical modification (edge creation) and chemical pathway (carbon insertion) for defect formation, and only in the specific scenario are SPEs reliably observed.

Sub-Section II: First, we describe how our model selection is guided by physical likelihood rooted in experimental observations. In the **Sub-Section I**, we have shown that the observed nanoindentation-induced SPEs are strongly correlated with carbon insertion. Therefore, this theoretical investigation focuses specifically on carbon-related defect models.

We carried out calculations on representative carbon-related defect configurations. As discussed in the Supplementary Information, among the various atomic configurations explored, certain two-carbon substitutional defects were found to exhibit electronic transition energies that closely match those single-photon emission observed experimentally. This alignment supports our following theoretical emphasis on two-carbon configurations embedded in zig-zag-type edge environments.

Secondly, we discuss how energetic favorability, evaluated via formation energies, further narrows down the most probable configurations in the nanoindentation context. We performed additional calculations of the formation energies of a series of carbon-related structures. The modeled structures, shown in Fig. R15a. include representative cases of carbon atoms embedded at either boron-terminated (B-edge) or nitrogen-terminated (N-edge) zig-zag-type edges, with configurations labelled CC1–CC3. For reference, pristine edge structures without carbon are also included.

The formation energies of various defects were calculated to evaluate their stability and the thermodynamic properties of defect formation during indentation. The formation energy is defined as:

$$E_f = E(D) - E(hBN) + \sum_i n_i (\mu_i + E_i)$$

and was computed under two extreme chemical potential conditions: nitrogen-rich and boron-rich limits (Fig. R15b and Fig.R15c), which represent two extreme cases of chemical potential during hBN growth or processing^[4]. Under nitrogen-rich conditions, boron vacancies or carbon substitutions at boron sites are energetically more favorable. And under boron-rich conditions, nitrogen-site substitutions or nitrogen vacancies are more favorable. These limits allow us to assess defect stability across a broad range of experimental environments. As shown in the bar plots, darker blue bars denote boron-edge-based configurations, and lighter blue bars represent nitrogen-edge-based ones.

From the figures, we observe that boron-terminated edges are generally more stable than nitrogen-terminated ones under both boron-rich and nitrogen-rich chemical environments, which is consistent with previous theoretical works. Additionally,

compared to isolated carbon substitutions, carbon dimers located at edge sites exhibit significantly lower formation energies. Among these, CC1 are especially favorable, likely due to local strain and under-coordination at geometrically confined sites, because corner sites are naturally strained and chemically more reactive due to their low coordination. These combined insights on physical likelihood and energetic stability indicate that carbon impurities preferentially form dimer defects at the corner sites of boron-terminated zig-zag-type edge.

Figure R15 (Figure 4 in the revised manuscript) | Formation energy of carbon-inserted edge defects in hBN. a, Atomic structures of zig-zag-type edge with and without carbon atoms under boron-edge and nitrogen-edge terminations (color coding: boron in green; nitrogen in grey; carbon in red). **b,** Calculated formation

energies under nitrogen-rich conditions. **c**, calculated formation energies under boron-rich conditions. Darker bars represent boron-edge-based configurations; lighter bars represent nitrogen-edge-based ones.

Sub-Section III: We acknowledge that standard single-particle DFT calculations, based on a single-particle approximation, have known limitations. In particular, they often suffer from delocalization errors, which may lead to inaccuracies in describing the localized electronic states associated with point defects in solids.

In principle, many-body correction methods—such as the GW approximation, which computes the electronic self-energy based on Hedin's equations using the Green's function (G) and the screened Coulomb interaction (W)—can yield improved quantitative accuracy. However, these approaches are computationally prohibitive for complex systems involving large supercells or for comprehensive comparison across a range of defect structures.

More computationally feasible approaches like quantum embedding have been developed to address these challenges.

Notably, in a recent work by one of our co-first-authors^[5], quantum embedding techniques were applied to examine the similar type of carbon-related defect structures in monolayer hBN. The study concluded that the many-body interactions do lead to shifts in defect energy levels, the qualitative features (defect configurations, emission trends, etc.) remain consistent with standard DFT predictions.

These findings support the validity of using single-particle DFT in our study. While we acknowledge that the quantitative limitations, the qualitative physical insights derived from DFT remain robust. Given the complexity of our system, we consider standard DFT a justified and tractable approach for screening defect candidates and interpreting experimental emission trends.

Action taken:

In the revised version of manuscript and SI, **we added Figure R3 - 4, Figure R15.**

In the revised version of manuscript, we added the following:

“To gain atomic-scale insight into the structure near nanoindentation sites, we employed aberration-corrected transmission electron microscope (AC-TEM, JEOL JEM-ARM300F Grand ARM) to investigate indented hBN flakes (See Supplementary Fig. S7). Figure 2e shows a fracture edge formed by direct contact with the nanotip. The nanoindentation process does not yield well-defined geometries, and we observe regions of varying thickness along the edge (multilayer region 1, monolayer region 2, and open hole region 3). At the fractured edge, highlighted by red dashed lines, zig-zag-type edges dominate. Sharp corner-like geometries, and triangular vacancy structures adjacent to edges are also observed.

We used the electron energy-loss spectroscopy (EELS) detector to probe fracture edge (Fig. 2e, Supplementary Fig. S7d and f), pristine area (Supplementary Fig. S7b) and folded edge (Supplementary Fig. S7c and e). This allowed us to examine the chemical composition of three different sites. The spectra are shown in Fig. 2f. As expected, the boron (~185 eV) and nitrogen (~400 eV) K-edges are observed in all regions. In contrast, a pronounced carbon K-edge signal (~285 eV) appears only at the fractured edge (red line). The carbon edge exhibits clear π^* and σ^* features, corresponding to transitions to unoccupied π and σ anti-bonding states, respectively, consistent with the presence of sp^2 -hybridized carbon^{51,52}. No carbon-related signals are detected in the spectra from the pristine hBN area (blue line) or folded edge (purple line), indicating that carbon incorporation is confined to regions that experienced direct mechanical contact with the carbon-coated tip. Atomic-scale

contrast analysis for elemental identification is performed using the inverse fast Fourier transform (iFFT) image derived from Fig. 2f (Supplementary Fig. S7), providing evidence for carbon atoms incorporated at edge sites.”

“As previously demonstrated, the formation of SPEs is strongly correlated with carbon insertion. Representative carbon-related defect configurations are illustrated in Fig. 4a, including carbon atoms embedded at either boron-terminated (B-edge) or nitrogen-terminated (N-edge) zig-zag-type edges, labelled CC1–CC3.

To evaluate defect stability, we computed the formation energy as:

$$E_f = E(D) - E(hBN) + \sum_i n_i (\mu_i + E_i)$$

under nitrogen-rich and boron-rich conditions (Fig. 4b and Fig.4c), representing two extreme cases of chemical potential during hBN growth or processing⁵⁷. The results show that carbon atoms decorated at the boron-terminated edges forming C - N bonds are more stable than at nitrogen-terminated edges, in agreement with previous theoretical works. Additionally, carbon dimers at edge or corner sites exhibit significantly lower formation energies than isolated carbon substitutions, likely due to local strain and under-coordination at geometrically confined sites. These combined insights on physical likelihood and energetic stability indicate that carbon impurities preferentially form dimer defects at the corner sites of boron-terminated zig-zag-type edge.”

Reference for Comment #3 of Reviewer #2

1. Huang, P., et al. "Carbon and vacancy centers in hexagonal boron nitride." *Physical Review B* 106.1 (2022): 014107.
2. Kim, Jaehyun, et al. "Electrochemically active porous carbon nanospheres prepared by inhibition of pyrolytic condensation of polymers." *Proceedings of the National Academy of Sciences* 120.19 (2023): e2222050120.
3. Toh, Chee-Tat, et al. "Synthesis and properties of free-standing monolayer amorphous carbon." *Nature* 577.7789 (2020): 199-203.
4. Huang, P., et al. "Carbon and vacancy centers in hexagonal boron nitride." *Physical Review B* 106.1 (2022): 014107.
5. Badrtdinov, Danis I., et al. "Dielectric environment sensitivity of carbon centers in hexagonal boron nitride." *Small* 19.41 (2023): 2300144.

Comment 4: One minor issue: Fig. 3(f) and the main text mention a saturation power of 172.4 microwatt, while the caption of Fig. 3 mentions 140.24 microwatt - maybe a misprint?

Response: We sincerely thank the Reviewer for pointing out this important issue. Indeed, this was a typographical error in the figure caption. We confirm that the correct saturation power is 172.4 μW , as stated in both the main text and the fitting result in Fig.3f.

The main text already provides the accurate description and fitted parameters:

“We measured the saturation behavior by increasing the excitation laser power to assess the SPE brightness. Considering the overall correction factors generated from our measurement (see more at Supplementary Note 2), Fig. 3f presents the emission intensity of SPE as a function of excitation laser power. The experimental

data (blue dots) were fitted using a conventional saturated emitter model: $I = I_{\infty} \times P/(P + P_{sat})$, where I_{∞} and P_{sat} represent the maximum achievable emission count rate and excitation power at the saturation excitation power, respectively.

The fitted red curve reveals a $I_{\infty} = 2.64$ MHz at $P_{sat} = 172.4$ μ W.”

We have corrected the Figure 3 caption accordingly as below to avoid confusion:

“**f**, Fluorescence saturation curve obtained from the single defect, showing the saturate power of 172.4 μ W. Inset: Excitation polarization recorded from the emitter. The solid red line is the fitting using a $\cos^2(\theta)$ function.”

Reply to Reviewer #1

Comment 1. The authors have satisfactorily addressed my concerns regarding the completeness of the experimental characterisation. They have provided the required high-resolution TEM data, additional PL results on temperature and thickness dependence, as well as clarifications of the simulation settings. The results and methodology are sound. Given the significance of this work and its potential benefit to the solid-state single-photon community, I recommend publication in Nature Communications.

Response: We sincerely thank the Reviewer for the positive evaluation of the completeness and significance of our study. We are also grateful for the Reviewer's recommendation for publication.

Reply to Reviewer #2

Comment 1. Concerning the revised manuscript by Luo et al., I believe that many of the questions and concerns have been (at least partially) met and answered, and the manuscript has improved.

In particular,

(1) the authors have carefully considered and summarized the various possibilities of why and how defect formation may take place, and

(2) the authors have with significant care specified their calculations and discuss their merits and limits. The discussion of formation energies is nice and helpful.

I fully understand that the current approach is a significant step towards (semi-)systematic generation of carbon-related defects in boron nitride, as an "engineering" procedure, and therefore warrants publication.

I am still a little bit (personally) disappointed that a fully systematic understanding of the formation can still not be given in the manuscript, simply because it is not possible at this point. However, this is clearly not the fault of the authors but results from the complexity of the situation.

Response: We gratefully thank the Reviewer for the positive and constructive evaluation of our revised manuscript. We are pleased that the Reviewer acknowledges our discussion of defect formation mechanisms and the clarification of our calculation methods. We fully agree with the Reviewer that a complete, systematic understanding of the formation process remains challenging due to the intrinsic complexity of the system. This topic will be further pursued in our future studies.